# Peculiar transcriptional reprogramming with functional impairment of dendritic cells upon exposure to transformed HTLV-1-infected cells

Auriane Carcone[1], Franck Mortreux[2], Sandrine Alais[1], Cyrille Mathieu[3], Chloé Journo[1ʘ], Hélène Dutartre [1ʘ]*

**1** Centre International de Recherche en Infectiologie, Retroviral Oncogenesis, Inserm U1111—Université Claude Bernard Lyon 1, CNRS, UMR5308, Ecole Normale Supérieure de Lyon, Université Lyon, Hospices Civiles de Lyon, Lyon, France, **2** Laboratory of Biology and Modelling of the Cell, University of Lyon, ENS de Lyon, University Claude Bernard, CNRS UMR 5239, Inserm U1210, Lyon, France, **3** Centre International de Recherche en Infectiologie, équipe Neuro-Invasion, TROpism and VIRal Encephalitis, Inserm U1111—Université Claude Bernard Lyon 1, CNRS, UMR5308, Ecole Normale Supérieure de Lyon, Université Lyon, Hospices Civiles de Lyon, Lyon, France

ʘ These authors contributed equally to this work.
* helene.dutartre@ens-lyon.fr

**Data Availability Statement:** All transcriptomic data were deposited on GEO under the accession

## Abstract

Manipulation of immune cell functions, independently of direct infection of these cells, emerges as a key process in viral pathophysiology. Chronic infection by Human T-cell Leukemia Virus type 1 (HTLV-1) is associated with immune dysfunctions, including misdirected responses of dendritic cells (DCs). Here, we interrogate the ability of transformed HTLV-1-infected T cells to manipulate human DC functions. We show that exposure to transformed HTLV-1-infected T cells induces a biased and peculiar transcriptional signature in monocyte-derived DCs, associated with an inefficient maturation and a poor responsiveness to subsequent stimulation by a TLR4 agonist. This poor responsiveness is also associated with a unique transcriptional landscape characterized by a set of genes whose expression is either conferred, impaired or abolished by HTLV-1 pre-exposure. Induction of this functional impairment requires several hours of coculture with transformed HTLV-1-infected cells, and associated mechanisms driven by viral capture, cell-cell contacts, and soluble mediators. Altogether, this cross-talk between infected T cells and DCs illustrate how HTLV-1 might co-opt communications between cells to induce a unique local tolerogenic immune microenvironment suitable for its own persistence.

## Author summary

Chronic viral infection is associated with an escape from immune surveillance. This may rely on the induction of inappropriate dendritic cells (DCs) responses, which can contribute to immunopathology. Immune dysfunctions have been repeatedly reported in people living with Human T-cell Leukemia Virus type 1 (HTLV-1), years before fatal clinical

number GSE266976. All other relevant data are in the manuscript and its supporting information files.

**Funding:** This study was supported by the Fondation pour la Recherche Médicale, DEQ. 20180339200, and by the Agence Nationale de la Recherche (ANR) to HD, ANR-22-CE15-0044. AC and CJ are employees of ENS de Lyon. CM is employee of CNRS. HD, FM and SA are employees of INSERM. The funders had no role in study design, data collection and analysis, decision to publish, or preparation of the manuscript.

**Competing interests:** The authors have declared that no competing interests exist.

symptom onset, including misdirected responses of DCs. Here, we report that HTLV-1-infected T cells actively manipulate neighboring, uninfected DC functions by rewiring their transcriptional response, leading to a biased, pro-tolerogenic responsiveness in DCs, induced by the bidirectional release of soluble mediators, in cooperation with mechanisms dependent on cell-cell contacts. This cross-talk illustrate how HTLV-1 might co-opt communications between cells to induce a local tolerogenic immune microenvironment suitable for its own persistence.

## Introduction

Chronic infection by the deltaretrovirus Human T-cell Leukemia Virus type 1 (HTLV-1) generally remains asymptomatic, but leads to severe diseases in 5 to 10% of infected individuals [1], in the form of Adult T-cell Leukemia/Lymphoma (ATL), or Tropical Spastic Paraparesis/HTLV-1 Associated Myelopathy (TSP/HAM). Remarkably, HTLV-1 chronic infection increases the risk of viral, bacterial or parasitic co-infections [2]. In agreement with these clinical observations, immune dysfunctions have been repeatedly reported in asymptomatic HTLV-1 carriers, years before severe diseases were diagnosed, indicating that chronic infection induces systemic effects and impairs the ability of the immune system to respond appropriately to infectious and non-infectious triggers [3–7]. Dendritic cells (DCs) are critical mediators of innate and adaptive immune responses during viral infection. Upon efficient viral recognition, DCs undergo a maturation program that renders them capable of inducing specific T-cell responses. The DC maturation process is characterized by a transcriptional program that translates, among others, in the upregulation of several maturation markers at the cell surface (*e.g.* CD80, CD86), accompanied by the secretion of pro-inflammatory cytokines such as tumor necrosis factor alpha (TNF-α), and antiviral cytokines such as type I interferons (IFN-I)[8]. However, inappropriate DC responses can contribute to immunopathology, a process that is exacerbated in chronic viral infections [9].

The role of DCs in HTLV-1 infection has been extensively investigated in terms of viral transmission to CD4$^+$ T cells, both *in vitro* [10–12] and in animal models [13,14]. Indeed, myeloid cells such as monocytes and DCs can harbor infectious HTLV-1 genomes in infected humans, including in asymptomatic individuals [6,15]. They exhibit impaired phenotypes and functions [15–17] that may correlate with their proviral load [18]. These clinical observations indicate that HTLV-1 could in fact dampen the responsiveness of infected DCs, which may ultimately contribute to immune dysfunction during chronic infection. Accordingly, DCs derived from ATL patients exhibit maturation defects [17] associated with an inability to induce proliferation of CD4$^+$ T cells [16]. However, because the proportion of DCs infected *in vivo* is low, the impact of HTLV-1 on DC function might not be solely attributed to HTLV-1-infected DCs, suggesting that the functions of non-infected DCs could be also modulated indirectly by HTLV-1 infection. In this work, we aimed at determining *in vitro* whether transformed HTLV-1-infected T cells could affect neighboring human DC functions in the absence of their infection. We show that upon coculture with HTLV-1-transformed T cell lines, human monocyte-derived DCs (MDDCs) do not fully mature. This impaired maturation is associated with a unique transcriptional signature in MDDCs, which differs from a typical maturation program, and is correlated with a dampened responsiveness of HTLV-1-exposed MDDCs to subsequent stimulation by TLR-4 or TLR-7 agonists. Interestingly, this tolerogenic state can also be induced by exposing freshly derived MDDCs for several hours to the supernatant from MDDCs cocultured with HTLV-1-infected T cells for several hours, highlighting a

cross-talk between MDDCs and T cells based on a bidirectional release of soluble mediators. This unique mechanism of viral-induced tolerance of DCs illustrates how HTLV-1 might co-opt communications between cells to induce a local immune microenvironment suitable for its own persistence.

## Results

### Human MDDCs do not fully mature when exposed to HTLV-1-infected T cells

To determine whether chronically infected T cells could directly or indirectly manipulate human DC functions *in vitro*, we first addressed whether infected T cells were sensed by DCs. We used human monocyte-derived DCs (MDDCs) cultured *in vitro* with HTLV-1-infected and transformed T cell lines, and assessed MDDCs maturation by flow cytometry after 24h or 48h of coculture, using the MDDC-specific CD11c marker to identify MDDCs in the coculture (see **Fig 1A** for the experimental settings, **and S1 Fig** for the gating strategy). At these time points, we have previously shown that less than 1% of MDDCs have integrated the viral genome but they do not release any virion [11]. In contrast to MDDCs exposed to measles virus (MeV), used as a pathogen known to induce MDCC maturation [19,20], MDDCs exposed to HTLV-1 transformed C91-PL cells, failed to fully upregulate CD86, a surrogate marker of DC maturation (**Fig 1B, left panel**). Repeated experiments using MDDC samples from independent donors showed that while around 90% of MDDCs upregulated CD86 upon exposure to MeV, regardless of being infected or not by MeV (**Fig 1B, right panel**, compare total versus GFP$^-$ and GFP$^+$ MDDCs), as well as upon stimulation by the TLR-4 agonist LPS (**Fig 1B, right panel**), the percentage of activated MDDCs after coculture with HTLV-1-transformed C91-PL cells remained low, slightly above Jurkat-exposed MDDCs but without statistical significance (**Fig 1B, right panel**). To exclude a cell line-specific effect, and to thoroughly characterize MDDCs maturation status, MDDCs were exposed to several control uninfected T cell lines (**S2A–S2F Fig**; Jurkat, CEM or Molt-4 T cell lines, green bars), as well as to several HTLV-1-transformed T cell lines (**S2A–S2F Fig**; C91-PL, MT-2 or Hut102; blue bars); and the regulation of both maturation markers CD86, CD83, CD80, and CD40, (**S2A–S2D Fig**) and inhibition markers ICOSL and PD-L1 (**S2E–S2F Fig**) at the cell surface was monitored by flow cytometry. All the HTLV-1-transformed T cell lines induced either no change, or only minor changes, in the expression (percentage of positive cells and normalized mean signal intensity, MFI) of maturation or inhibition markers at the surface of cocultured MDDCs (**S2A–S2F Fig**) even after sustained coculture for 48 h (**S2A–S2F Fig**), excluding a possible delay in maturation. The inefficient upregulation of MDDC maturation markers was also accompanied by a low ability to secrete TNF-α or IFN-I (**Figs 1C** and **S2G**). Thus, HTLV-1-infected T cells are unable to induce a typical maturation program in MDDCs, as if they were not efficiently sensed by human DCs. Since similar observations were made with all HTLV-1-infected T cell lines tested, and with the tested maturation markers and cytokines, the following experiments were done using the C91-PL T cell line and CD86 as DC maturation marker.

From our previous work, MDDCs efficiently capture HTLV-1 after coculture with HTLV-1-infected T cell lines [11], suggesting that in our experimental conditions, HTLV-1 capture itself is inefficient in driving MDDC maturation. To formally demonstrate this, we assessed HTLV-1 capture in MDDCs following coculture, by staining the HTLV-1 Gag p19 structural protein. As expected, MDDCs were efficient at capturing HTLV-1 (**Fig 1D**), irrespective of the infected T cell line used in the coculture (**S2H Fig**), with around 30% of capture observed in a representative coculture experiment with C91-PL cells (**Fig 1D, left**), and reaching up to 80% of capture observed in repeated experiments using independent donors (**Fig 1D, right**).

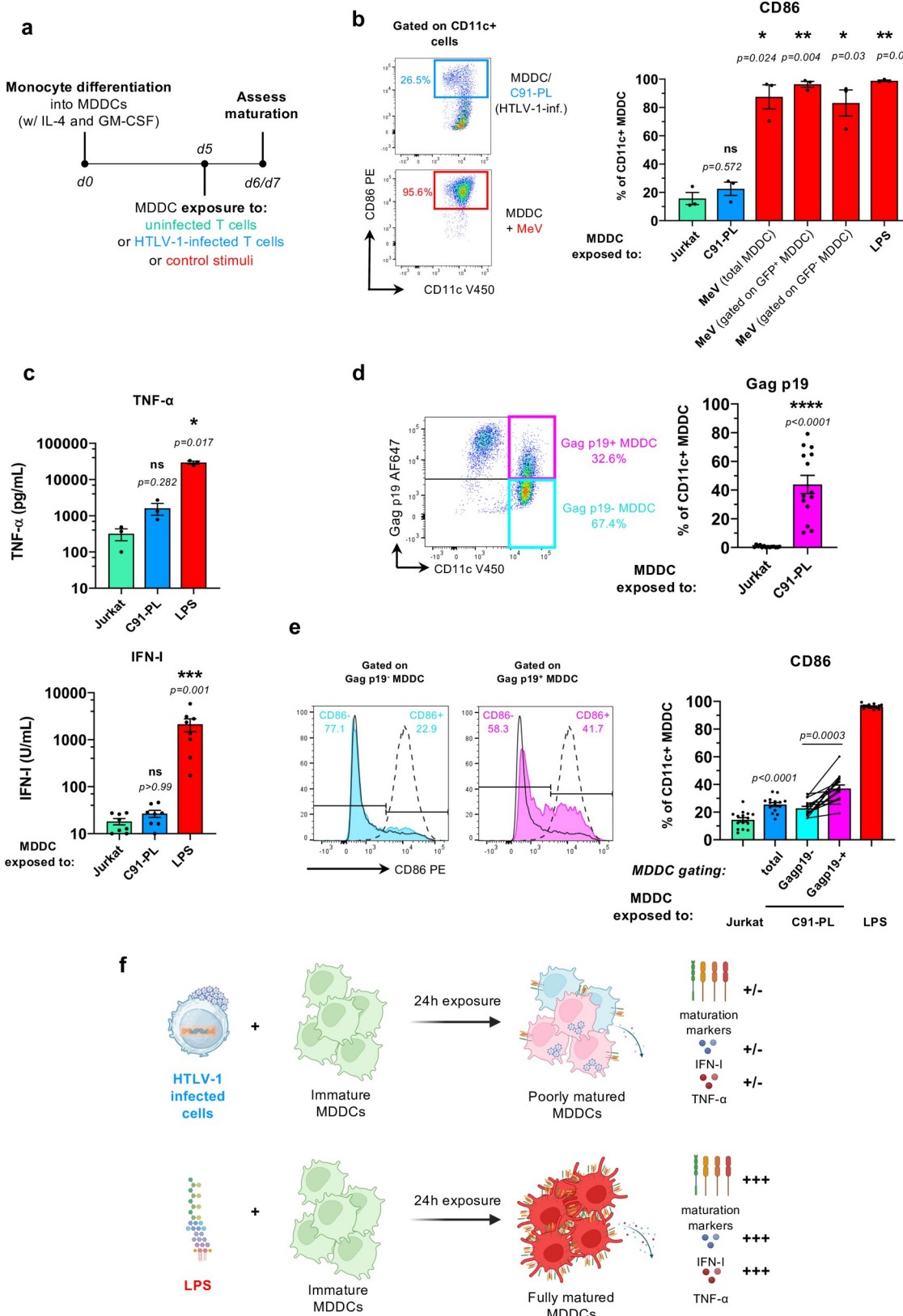

**Fig 1. Human MDDCs do not fully mature when exposed to HTLV-1-infected T cells. a.** Schematic representation of the experimental design. From day 0 (d0), monocytes were differentiated into MDDCs by a 5-day culture in the presence of IL-4 and GM-CSF. At day 5 (d5), MDDCs were cocultured with control uninfected or HTLV-1-infected T cell lines for 24h (or 48h, see

S2A–S2F Fig). Alternatively, MDDCs were stimulated with GFP-expressing MeV or LPS (control stimuli) for 24h (or 48h, see S2A–S2F Fig). Cells and culture supernatants were collected at d6 (or d7, see S2A–S2F Fig). **b.** Flow cytometry analysis at d6 after CD11c and CD86 staining. The percentage of CD86$^+$ MDDCs (indicated gate) among total CD11c$^+$ MDDCs was quantified. **Left**: representative experiment. **Right**: results from n = 3 independent experiments, analyzed with RM one-way ANOVA using MDDCs exposed to Jurkat as a basis, as described in S7 Table. **c.** Supernatant from the indicated cocultures or from LPS-stimulated MDDCs was collected at d6, and TNF-α (**top**) and IFN-I (**bottom**) were quantified by Luminex and reporter cell assay, respectively. Results from n = 3 or 8 independent experiments, respectively. Presented data are a subset of S2G Fig analyzed with RM one-way ANOVA (top) or Kruskal-Wallis test (bottom) using MDDCs exposed to Jurkat as a basis, as described in S7 and S8 Tables. **d.** Viral capture in MDDCs cocultured with Jurkat or C91-PL T cells was assessed at d6 by Gag p19 staining on CD11c$^+$ MDDCs. The percentage of Gag p19$^+$ and Gag p19$^-$ MDDCs among total CD11c$^+$ MDDCs was determined. **Left**: representative experiment. **Right**: results from n = 14 independent experiments analyzed with Kruskal-Wallis test as described in S7 and S8 Tables as a subset of S2H Fig. **e.** Flow cytometry analysis after CD11c, Gag p19 and CD86 staining. **Left**: representative experiment. Dashed lines: LPS-exposed MDDCs. Solid black lines: Jurkat-exposed MDDCs, light blue line: Gag p19$^-$ MDDCs. Magenta lines: Gag p19$^+$ MDDCs. **Right**: the same color code is used to represent results from n = 15 independent experiments analyzed with one-way ANOVA using MDDCs exposed to Jurkat as a basis, as described in S7 Table. **f.** Schematic drawing summarizing the results from Figs 1 and S2. The drawing was created using BioRender. com. Note that statistical analysis using C91-PL-exposed MDDCs as a basis gives similar results and is also presented in S7 Table.

Maturation was then compared in MDDCs that had not captured HTLV-1 (Gag p19-negative MDDCs, cyan, **Fig 1E**), or had captured HTLV-1 (Gag p19-positive MDDCs, magenta, **Fig 1E**). Note that in this experiment, the difference in CD86 expression between Jurkat-exposed and C91-PL-exposed MDDCs (total population) did reach statistical significance (green and blue bars, **Fig 1E**, right), probably due to a higher number of replicates compared to **Fig 1B**, although this difference remained very limited compared to LPS-induced maturation (see **S7 Table** for statistical analysis using C91-PL-exposed MDDCs as a basis for multiple comparisons). In addition, although CD86 expression was consistently higher in p19-positive MDDCs (**Fig 1E**, left, compare the two histograms and **right**, compare cyan and magenta bars), the percentage of CD86-positive Gag p19-positive MDDCs remained low compared to fully matured LPS-exposed MDDCs (**Fig 1E**, right compare magenta and red bars), highlighting that maturation was restricted even in MDDCs that had captured HTLV-1. No significant correlation was detected among repeated experiments between the level of HTLV-1 capture and the efficiency of MDDC maturation (**S2I Fig**, p = 0.1412), further confirming that HTLV-1 capture itself is inefficient in driving MDDC maturation.

Taken together, these observations, recapitulated in **Fig 1F**, demonstrate that cell-associated HTLV-1 fails to fully mature MDDCs, suggesting that it is poorly sensed by these human innate cells.

## Exposure to HTLV-1-infected T cells induces a unique transcriptional signature in MDDCs

In contrast to MDDCs, we previously showed that plasmacytoid dendritic cells (pDC) do efficiently sense cell-associated HTLV-1[21]. Inefficient sensing of HTLV-1-infected T cells by MDDCs could then be the result of either a stealth behavior of cell-associated virus towards MDDCs, resulting in a *bona fide* lack of response specifically in this cell type, or, alternatively, to the active manipulation of MDDC functions by HTLV-1-infected T cells, resulting in a specific response that differs from a typical maturation program. To discriminate between both, we aimed at determining the complete transcriptomic landscape of MDDCs after exposure to HTLV-1-infected T cells, or to control uninfected T cells, using bulk RNA-seq (**Fig 2**). After 24h of coculture, MDDCs were magnetically separated from C91-PL or control Jurkat T cells, based on the exclusive expression of CADM1 on T cells (**S3A Fig**). Note that although the levels of CADM1 were higher at the surface of C91-PL cells compared to Jurkat cells (**S3A Fig**), a similar MDDC enrichment yield of around 99% was observed after separation in both conditions, with no significant T cell contamination (**S3B Fig**). LPS-stimulated MDDCs were also

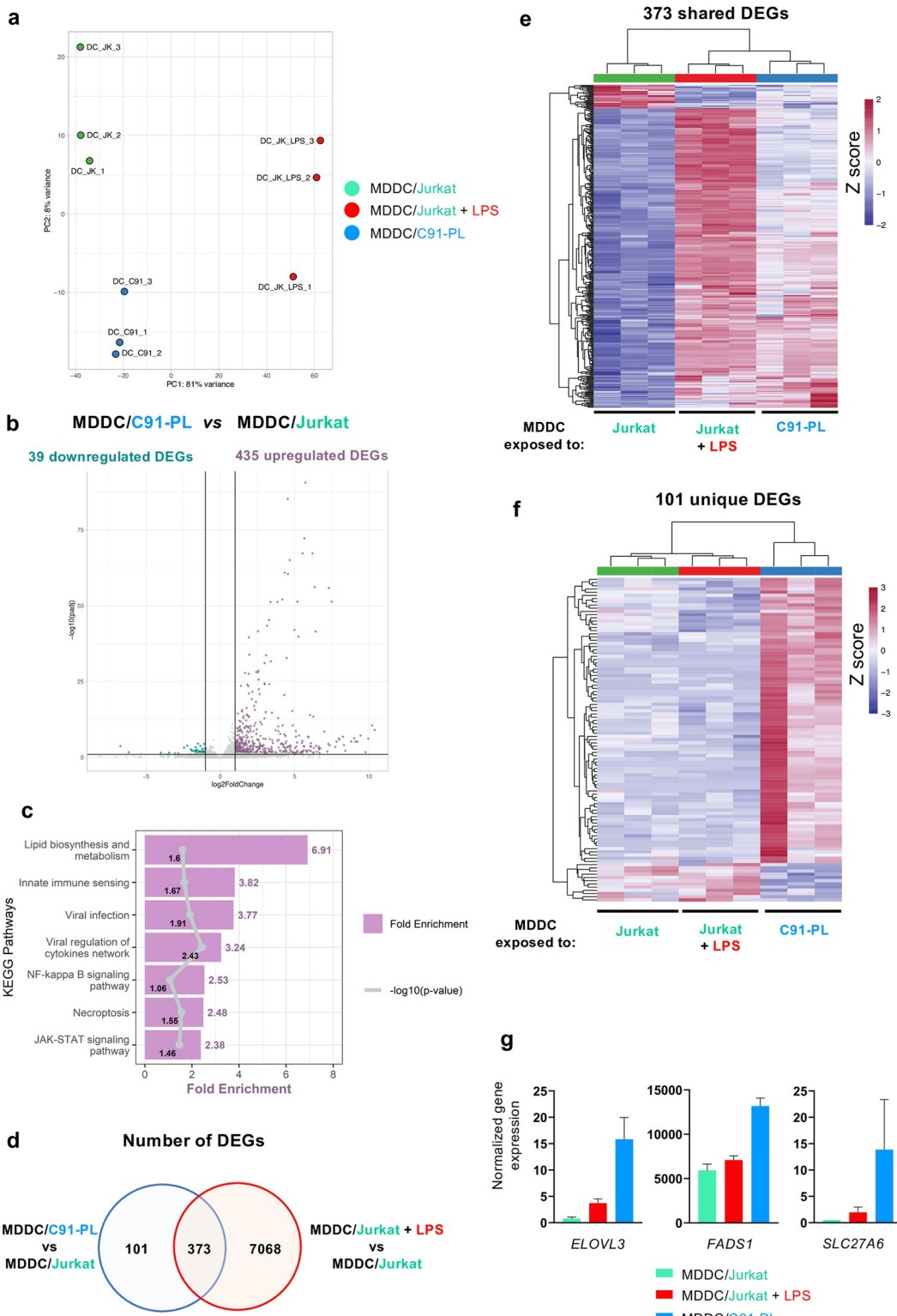

**Fig 2. Exposure to HTLV-1-infected T cells induces a unique transcriptional signature in MDDCs.** MDDCs were cocultured for 24h with control Jurkat cells, or with HTLV-1-infected C91-PL cells, or with control Jurkat cells in the presence of LPS. Cells were then separated and RNA-seq analysis was performed on MDDCs in 3 independent replicates. **a.** Principal component analysis of global similarities in gene expression between samples (DC_ JK: MDDCs after coculture with Jurkat cells, green; DC_JK_LPS: MDDCs after coculture with Jurkat cells in the presence of LPS, red; DC_C91-PL:

MDDCs after coculture with C91-PL cells, blue). **b.** Volcano plot of DEGs between MDDCs exposed to C91-PL and MDDCs exposed to Jurkat cells. **c.** KEGG pathways analysis performed on the 435 upregulated DEGs. Bars represent the fold-enrichments, while dots represent the p-values expressed as log10 for each pathway, or their respective mean for grouped pathways (see S2 Table). **d.** Venn diagram of the DEGs in C91-PL-exposed MDDCs compared to Jurkat-exposed MDDCs (blue circle), and of the LPS-modulated DEGs in Jurkat-exposed MDDCs (red circle). **e-f.** Heatmap of the 373 shared (**e**) and 101 unique (**f**) DEGs. Gene expression was normalized per row. **g.** DESeq2 normalized counts for *ELOVL3*, *FADS1*, and *SLC27A6* across samples.

included in the RNA-seq analysis, as a reference for fully matured MDDCs. RNA was extracted from each MDDC sample, and submitted to quality control followed by sequencing.

First, to show global similarities in gene expression between samples, unsupervised hierarchical clustering and principal component analysis (PCA) were performed on normalized transcript count tables. Analysis of sample clustering along the two first PCs (**Fig 2A,** PC1 and PC2), that explain 81 and 8% of the total variance, respectively, showed that the main source of variance in the dataset was the stimulation by LPS, followed by the infection status of cocultured T cells, while the identity of the MDDC donor contributed only weakly to the global variance. In agreement with flow cytometry data shown in Fig 1, this confirmed, at the transcriptional level, the inefficient activation of MDDCs after exposure to HTLV-1-infected T cells, which contrasts with the typical maturation program of fully activated MDDCs. This also indicates that exposure to HTLV-1-infected T cells does induce a detectable, yet subtle, transcriptional response, when compared to exposure to uninfected T cells.

To characterize the changes in gene expression in MDDCs induced by exposure to HTLV-1-infected T cells, we used DESeq2 to identify differentially expressed genes (DEGs) between C91-PL- and Jurkat-exposed MDDCs. As a reference for a typical maturation program, we identified DEGs between LPS- and Jurkat-exposed MDDCs. Using a fold-change cut-off of 2, a total number of 474 DEGs were obtained between C91-PL- and Jurkat-exposed MDDCs (padj<0.05), with 435 genes found significantly upregulated upon HTLV-1 exposure, and 39 genes found significantly downregulated (**Fig 2B** and **S1 Table**). A heatmap of the 474 DEGs showed clustering between samples, as expected (**S4A Fig**).

To gain insight into the processes modulated in MDDCs after exposure to HTLV-1-infected T cells, we performed gene ontology analysis on the set of 435 upregulated genes. Over-represented KEGG pathways were retrieved and filtered to reduce redundancy (see **S2 Table**), leading to a final list of 7 pathways (**Fig 2C** and **S2 and S3 Tables**). Strikingly, exposure to HTLV-1-infected T cells was characterized by significant over-representation of upregulated genes involved in lipid biosynthesis and metabolism (**Fig 2C**). Interestingly, genes involved in several pathways linked with response to viral infection were also over-represented, including genes involved in innate immune sensing (sub-pathways: RIG-I or Toll-like receptors and NOD-like receptors), in viral infection (sub-pathways: Influenza A, Measles, Hepatitis C, Hepatitis B, Human Papillomavirus, Herpes simplex virus 1, Coronavirus, HIV, EBV), and in the NF-κB signaling pathway. Taken together, these results confirm that exposure to HTLV-1-infected T cells does induce a transcriptional response in MDDCs, indicating that sensing does occur to some extent, but does not culminate in a typical maturation and anti-viral program.

To further determine the extent to which this transcriptional response is distinct from a typical maturation signature, we compared the genes affected by LPS stimulation of Jurkat-exposed MDDC to those affected by MDDC exposure to C91-PL. Among the 474 DEGs, 373 genes were also differentially expressed after LPS stimulation (**Fig 2D**), representing only 5% of all LPS-modulated genes. Among these shared genes, 7 genes annotated as involved in the NF-κB signaling pathway were retrieved, including *CCL19*, *TNFSF13B*, *TRIM25*, *BCL2L1* and *BCL2A1* (**S4B Fig**). However, the level of up-regulation of these shared NF-κB-related genes was strikingly lower after HTLV-1 exposure compared to LPS stimulation (**S4B Fig**). This

observation of a lower magnitude of differential expression (either up- or downregulation) was general to most of the 373 shared DEGs (**Fig 2E**). Of note, the *RELA*, *RELB* and *NFKB1* genes that were upregulated by LPS stimulation were not upregulated upon C91-PL exposure (**S4C Fig**), possibly contributing to the lower activation of the NF-κB-dependent genes observed above (**S4B Fig**). In addition, among the 372 Interferon-Stimulated Genes (ISG) listed in **S1 Table**, only 121 were retrieved in the set of significantly upregulated genes, and again, their up-regulation was lower than that induced by LPS stimulation (**S4D Fig**). These observations strengthen the notion that while sensing of HTLV-1-infected T cells does occur to some extent, the magnitude of the maturation and antiviral transcriptional response remains very limited, which is consistent with the inefficient maturation of MDDCs observed at the protein expression level (see **Fig 1**).

Interestingly, among the 474 DEGs, 101 genes were not present in the list of DEGs between LPS- and Jurkat-exposed MDDCs (**Fig 2D and 2F**), defining a unique transcriptional signature associated with the exposure to HTLV-1-infected C91-PL cells. Due to the limited number of genes included in this specific signature, gene ontology analysis was not feasible. Interestingly however, several genes retrieved in Fig 2C as those involved in lipid biosynthesis and metabolism, were also included in this specific signature, such as *ELOVL3* (Elongation of Very Long Fatty Acid Elongase 3), *FADS1* (Fatty Acid Desaturase 1), and *SLC27A6* (Solute Carrier Family 27 Member 6, a member of the fatty acid transport protein family) (**Fig 2G**). Altogether, these transcriptomic analyses show that exposure to HTLV-1-infected T cells is not completely silent in MDDCs, but rather results in a unique transcriptional response that differs from a typical maturation program. This supports the notion that HTLV-1-infected T cells might actively manipulate MDDC functions to limit their functional response, possibly by rewiring lipid biosynthesis and metabolism.

### Pre-exposure to HTLV-1-infected T cells dampens the responsiveness of MDDCs to subsequent stimulation

We next aimed at investigating whether this unique transcriptional response observed upon exposure to HTLV-1-infected T cells indeed translates into functional defects in MDDCs, beyond their inefficient maturation. To this end, we addressed whether pre-exposure to HTLV-1-infected cells influenced MDDC responsiveness when exposed to a subsequent stimulation. After 24h of coculture with HTLV-1-infected T cell lines, MDDCs were restimulated with strong inducers of MDDCs maturation, in the form of LPS (TLR-4 ligand) or R848 (TLR-7/8 ligand, **Fig 3A**), for another 24 h. When compared to MDDCs pre-exposed to uninfected control T cell lines (red bars), MDDCs pre-exposed to HTLV-1-transformed T cells (dark blue bars) upregulated CD86 (as well as other maturation markers) to a significantly lower extent when stimulated by LPS or R848 (**Figs 3B and S5A–S5D**). In addition, TNF-α secretion by MDDCs upon LPS stimulation was also significantly reduced by pre-exposure to HTLV-1-transformed T cells (**Figs 3C**, left panel, and **S5E**), while IFN-I secretion was not affected (**Figs 3C**, right panel, **and S5E**). These results indicate that pre-exposure to HTLV-1-transformed T cells indeed influences the responsiveness of MDDCs to subsequent stimulation, by specifically dampening their pro-inflammatory response (monitored here through the upregulation of maturation markers and through TNF-α secretion), without hampering their antiviral response (monitored here through IFN-I secretion) (recapitulated in **Fig 3D**). This suggests that HTLV-1 might specifically manipulate the responsiveness of certain signaling pathways in MDDCs, such as the NF-κB pathway upstream of the pro-inflammatory program, while leaving the responsiveness of others unaffected, such as the IRF3 pathway upstream of the antiviral program.

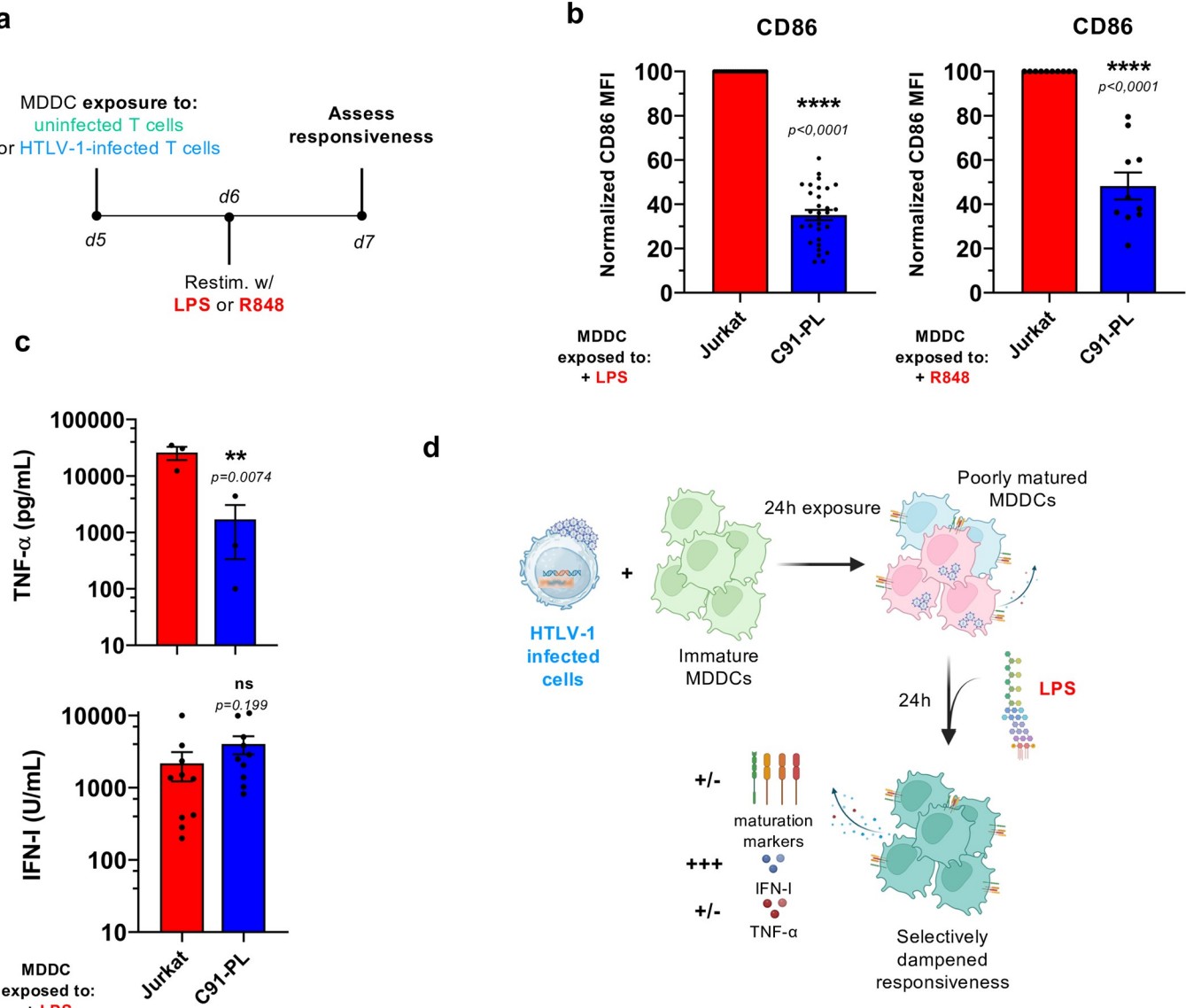

**Fig 3. Pre-exposure to HTLV-1-infected T cells dampens the responsiveness of MDDCs to subsequent stimulation. a.** Schematic representation of the experimental design. MDDCs were exposed to uninfected (Jurkat, green) or HTLV-1-infected T cells (C91-PL, blue) for 24h, before restimulation with LPS or R848 for an additional 24h. **b.** Flow cytometry analysis after CD11c and CD86 staining, in LPS- (**left**) or R848-restimulated MDDCs (**right**) pre-exposed to Jurkat (red) or C91-PL (dark blue) cells. Data are represented as the normalized MFI of CD86, with the MFI in restimulated Jurkat-pre-exposed MDDCs set to 100 (n = 30 or 10 independent experiments, respectively). Presented data are a subset of S5A or S5D Fig, respectively and were analysed with Kruskal-Wallis test or ordinary one-way ANOVA, respectively as described in S7 and S8 Tables **c.** Supernatant from the indicated cocultures was collected after LPS restimulation, and TNF-α (**left**) and IFN-I (**right**) concentrations were quantified for n = 3 or 10 independent experiments, respectively. Presented data are a subset of S5E Fig and were analysed with RM one-way ANOVA or ordinary one-way ANOVA, respectively as described in S7 and S8 Tables. **d.** Schematic drawing summarizing the results from Figs 2 and S5. The drawing was created using BioRender.com.

## Pre-exposure to HTLV-1-transformed T cells alters the transcriptional response of MDDCs to subsequent stimulation

To obtain a broader overview of how pre-exposure to HTLV-1-transformed T cells affects the responsiveness of MDDCs, a second bulk RNA-seq analysis was conducted on MDDCs pre-exposed to an HTLV-1-infected T cell line (C91-PL), or to a control uninfected T cell line (Jurkat), and then re-stimulated by LPS (**Fig 4**). To address whether viral capture was required for

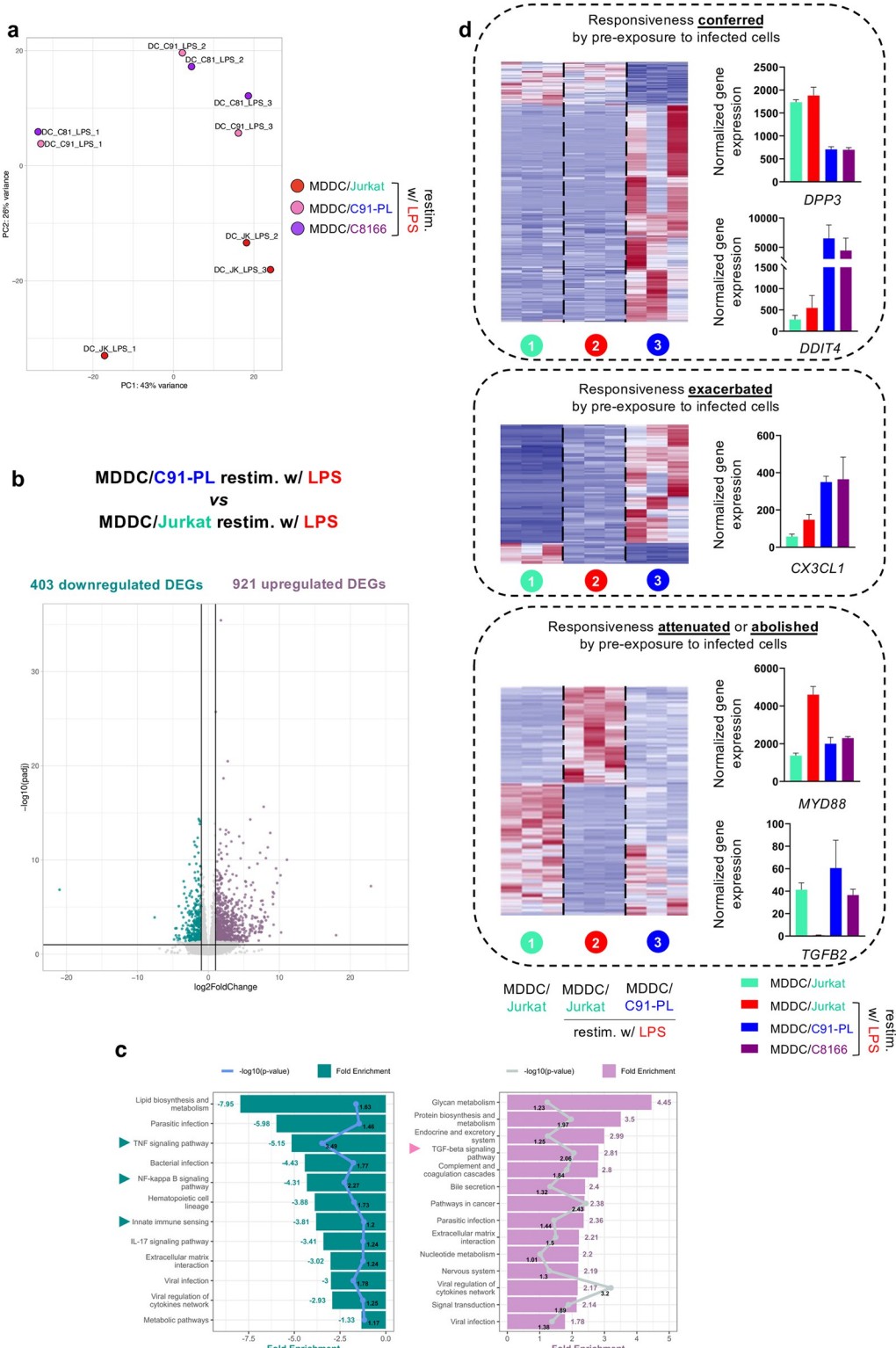

**Fig 4. Pre-exposure to HTLV-1-infected T cells alters the transcriptional response of MDDCs to subsequent stimulation.** MDDCs were cocultured for 24h with Jurkat cells, or with HTLV-1-infected C91-PL or C8166 cells, before restimulation with LPS for an additional 24h. Cells were then separated and RNA-seq analysis was performed on MDDCs in 3 independent replicates. **a.** Principal component analysis of global similarities in gene expression between samples (DC_JK_LPS: Jurkat-pre-exposed MDDCs after LPS restimulation, red; DC_C91_LPS: C91-PL-pre-exposed

MDDCs after LPS restimulation, pink, DC_C81_LPS: C8166-pre-exposed MDDCs after LPS restimulation, purple). **b.** Volcano plot of DEGs between C91-PL-pre-exposed MDDCs after LPS restimulation, and Jurkat-pre-exposed MDDCs after LPS restimulation. **c.** KEGG pathways analysis performed on the 403 downregulated DEGs (**left**) or on the 921 upregulated DEGs (**right**). Bars represent the fold-enrichments, while dots represent the p-values expressed as log10 for each pathway, or their respective mean for grouped pathways. **d.** Heatmaps and DESeq2 normalized counts across samples of illustrative genes, showing expression across samples of genes belonging to the "conferred" (**top**), "exacerbated" (**middle**) or "attenuated or abolished" (**bottom**) responsiveness pattern.

this manipulation of MDDC responsiveness, we also included in the RNA-seq analysis MDDCs cocultured with the HTLV-1-infected C8166 T cell line that does not produce viral particles [22]. As expected, no viral particle was captured by C8166-exposed MDDCs, in contrast to C91-PL-exposed MDDCs (**S6 Fig**).

PCA analysis of samples showed that both the donor and the infection status of cocultured T cells were the main sources of variance in the dataset (**Fig 4A**), confirming that pre-exposure to HTLV-1-transformed T cells does alter the transcriptional response of MDDCs to subsequent stimulation. In contrast, the ability to capture HTLV-1 did not contribute to a visible extent to the global variance, suggesting that viral capture might not be required for this alteration.

We then retrieved the lists of genes differentially expressed between the different experimental conditions. In agreement with the PCA analysis, 1324 genes were found differentially expressed upon LPS stimulation between C91-PL-pre-exposed MDDCs, and Jurkat-pre-exposed MDDCs (**S4 Table**), with a total of 403 DEGs being downregulated and 921 DEGs being upregulated (**Fig 4B**). Gene ontology analysis was conducted on these sets of downregulated and upregulated genes, respectively (**Fig 4C**, left and right panels, respectively). The set of downregulated genes was characterized by a significant over-representation of genes annotated as being involved in TNF-α or NF-κB signaling and innate immune sensing (**Fig 4C**, left panel, arrows, see **S5 Table**).

Conversely, the set of upregulated genes was characterized by a significant over-representation of genes annotated as being involved in the TGF-β signaling pathway (**Fig 4C**, right panel, arrow, see **S6 Table**). This confirmed that pre-exposure to HTLV-1-transformed T cell does influence the transcriptional response to LPS stimulation. To further identify the expression patterns of these genes differentially expressed between C91-PL-pre-exposed MDDCs, and Jurkat-pre-exposed MDDCs upon LPS stimulation, we classified these genes based on their differential expression across conditions (detailed in **S7A Fig** **and** see also **S4 Table**) as follows: (i) Genes whose responsiveness to LPS (be it repression or induction) is specifically conferred by pre-exposure to infected cells (**Fig 4D**, upper panel). (ii) Genes whose responsiveness to LPS (be it induction or repression) is exacerbated by pre-exposure to infected cells (**Fig 4D**, middle panel). (iii) Genes whose responsiveness to LPS is attenuated or (iv) abolished by pre-exposure to infected cells (**Fig 4D**, lower panel). This classification reveals that pre-exposure to HTLV-1-transformed T cells induces both a change in the identity of genes that transcriptionally respond to LPS stimulation, defining a unique LPS-induced transcriptional signature; and a change in the magnitude of the transcriptional response of genes that are normally responsive to LPS.

In line with the gene ontology analysis presented in **Fig 4C**, the responsiveness of genes involved in NF-κB signaling, such as *MYD88* (**Fig 4D**, lower panel, see graph), or the proinflammatory genes *CCL2*, *IL12B*, *TNF*, *CXCL10* and *CXCL11* (**S7B Fig**, upper panel), was found drastically (around 10x) attenuated by pre-exposure to infected cells (C91-PL or C8166). This could account for the inefficient maturation and production of pro-inflammatory cytokines by HTLV-1-pre-exposed MDDCs after LPS stimulation, which was observed in Fig 3. In addition, in line with the gene ontology analysis presented in **Fig 4C**, genes of the

TGF-β signaling pathway, including *BMP6*, *BMP7*, and *IL13* (**S7B Fig**, middle panel), which do not respond to LPS in normal conditions, were found to be responsive to LPS when MDDCs were pre-exposed to HTLV-1-transformed cells, while the expression of *TGFB2*, which is repressed upon LPS stimulation in normal conditions, was maintained by pre-exposure to HTLV-1-infected cells (**Fig 4D**, lower panel, see graph). Interestingly, these genes encode cytokines known to participate in the induction of a tolerogenic immune microenvironment, with Treg and T$_H$2 responses, which are inefficient at controlling viral infection. Finally, DCs pre-exposed to HTLV-1-transformed cells strongly expressed *DDIT4* upon LPS stimulation (**Fig 4D**, upper panel, see graph). This gene has been reported in other contexts of tolerogenicity [23]. Of note, and in agreement with the efficient induction of IFN-I after LPS stimulation in both Jurkat- or C91-PL-pre-exposed MDDCs (see Fig 3C), the induction level of *ISGs* such as *ISG15*, *IFI44* and *IRF2* was not affected by pre-exposure to HTLV-1-infected cells (**S7B Fig**, lower panel).

Altogether, this transcriptomic analysis indicated that pre-exposure to HTLV-1-transformed T cells influences the responsiveness of MDDCs to subsequent stimulation, by specifically dampening their pro-inflammatory response at the transcriptional level. In addition, it uncovered the fact that pre-exposure to HTLV-1-transformed T cells allows a unique set of genes to respond to LPS stimulation, triggering a biased, pro-tolerogenic response of MDDCs upon subsequent stimulation.

## Neither HTLV-1 viral capture, nor cell-cell contact with infected T cells, are strictly required to dampen the responsiveness of MDDCs to subsequent stimulation

As stated above, RNA-seq analysis using the "non virus producer" C8166 infected cell line suggests that viral capture might not be required to alter the transcriptional response of MDDCs to a secondary stimulation. To confirm this notion at the functional level, we repeated the coculture experiment followed by LPS stimulation, and analyzed MDDC maturation profile by flow cytometry (**Fig 5A**). Despite the lack of viral particle capture by C8166-exposed MDDCs (see **S6 Fig**), C8166 pre-exposure still dampened the responsiveness of MDDCs to LPS and R848 stimulation, as monitored through CD86, CD83 or CD80 upregulation, similar to C91-PL pre-exposure (**Figs 5A and S8A**), confirming that viral capture is not strictly required. Although C8166 do not produce viral particles, they might still engage in cell-cell contacts with MDDCs [24], which could be the trigger of the dampening of MDDC responsiveness. We addressed this hypothesis by performing the cocultures in transwells, in which MDDCs were physically separated from infected T cells by a permeable membrane (**Fig 5B**). Of note, reduced but detectable levels of viral capture were detected in MDDCs physically separated from C91-PL (**S8B Fig**), most probably because the virus is preferentially associated to the cell surface of infected cells and could be released as large adhesive viral aggregates [25] poorly able to cross the 0.4μm pores of the permeable membrane. The absence of physical contact between MDDCs and infected T cells, either producing (C91-PL) or not producing viral particles (C8166), did not restore a fully efficient maturation after LPS stimulation (**Fig 5B**), indicating that cell-cell contacts are not strictly required to allow HTLV-1-infected T cells to manipulate MDDC responsiveness. Of note however, C8166 cells appeared less efficient than C91PL cells in dampening MDDC responsiveness upon transwell coculture. This could result from additive effects of mechanisms dependent on viral capture on the one hand, and of cell-cell contact on the other hand: indeed, in the presence of cell-cell contacts (**Fig 5A**), the contribution of viral capture might be negligible; while in the absence of cell-cell contacts (**Fig 5B**), such contribution would become detectable. Consistently, comparison of MDDC

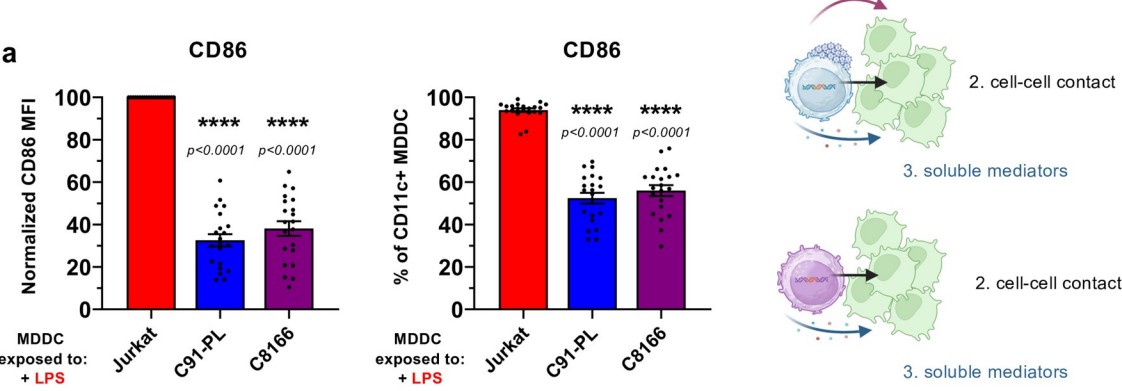

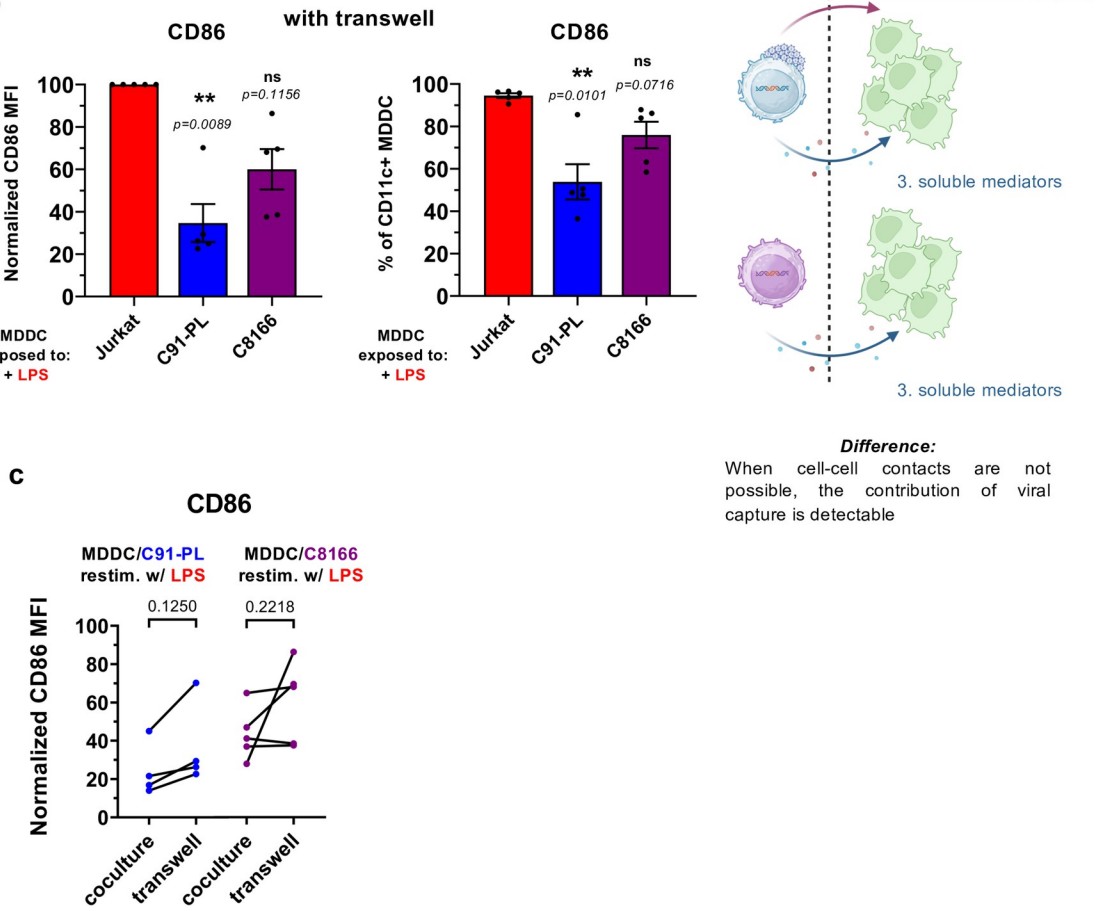

**Fig 5. Neither HTLV-1 viral capture, nor cell-cell contact with infected T cells, are strictly required to dampen the responsiveness of MDDCs to subsequent stimulation. a-b.** MDDCs were cocultured for 24h with Jurkat cells, or with HTLV-1-infected C91-PL or C8166 cells, in the absence (**a**) or presence (**b**) of a transwell insert (0.4μm pore diameter), before re-stimulation with LPS for an additional 24h. Flow cytometry analysis after CD11c and CD86 staining. Data are represented as the normalized MFI of CD86, with the MFI in re-stimulated Jurkat-exposed MDDCs set to 100 (**left**), or as the percentage of CD86⁺ MDDCs (**right**), and

summarize n = 21 (**a**) or n = 5 (**b**) independent experiments. Red: Jurkat-exposed MDDCs restimulated with LPS; dark blue: C91-PL pre-exposed MDDCs restimulated with LPS; purple: C8166 pre-exposed MDDCs restimulated with LPS. Detailed statistical analysis is presented Briefly, the statistical difference between means was tested using RM one-way ANOVA (a, left and b, right) or Friedman test (a, right and b, left), in accordance with the distribution. **c.** Paired percentage of CD86+ MDDCs in standard coculture (as measured in **a**), or with a transwell insert (as measured in **b**), with HTLV-1-infected C91-PL (**left**) or C8166 cells (**right**). Data summarized n = 4 or 5 independent experiments, respectively analysed with paired t-test or its non-parametric equivalent Wilcoxon matched-pairs signed rank test as described in S7 Table. **d.** Schematic drawing summarizing the results from Figs 5 and S8. The drawing was created using BioRender.com.

responsiveness in paired experiments of MDDC pre-exposed to infected cells with (coculture) or without (transwell) physical contacts showed that preventing physical contacts between MDDCs and infected cells resulted in a slightly higher, yet not significantly different, MDDC responsiveness (**Fig 5C**). This indicates that viral capture and/or cell-cell contacts may participate, but are not strictly required to allow HTLV-1-infected T cells to manipulate MDDC responsiveness, and suggests the contribution of a distantly acting set of soluble mediators.

### Impairment of MDDCs is partially recapitulated by their culture in conditioned medium from HTLV-1-infected T cells and MDDCs co-culture

Since neither viral particles nor cell-cell contacts are strictly required to manipulate MDDC responsiveness, we then tested the conditioning ability of the supernatant of HTLV-1-infected T cells. More specifically, we hypothesized that IL-10 produced by HTLV-infected T cells [26] could be a candidate mediator, as it was reported to have tolerogenic properties [27]. However, except for the supernatant of the MT-2 infected T cell line, IL-10 was not detected in the supernatant of any of the other infected T cell lines (**S9A Fig**). Accordingly, none of these supernatants were able to prevent MDDC responsiveness to LPS stimulation (**S9B Fig**). Because of the indication of the involvement of soluble mediators (see **Fig 5** above) we speculated that such mediators could be specifically released upon the co-culture between HTLV-1-transformed T cells and MDDCs. To test this hypothesis, we collected the conditioned medium of MDDCs cocultured with infected T cells for 24h, and used it to culture fresh, naïve MDDCs derived from the same monocyte donor (autologous cells) for 24h, before LPS stimulation for another 24h (see **Fig 6A** for the experimental design). In contrast to the supernatant of infected T cells alone (**S9B Fig**), the supernatant from the co-culture was still able to dampen the responsiveness of MDDCs (**Fig 6B**), although with less potency compared to coculture with infected T cells, as observed by analysis of paired experiments (**Fig 6C**). This is consistent with the tendency observed upon transwell cultures (see **Fig 5C**), and confirms the release of soluble tolerogenic mediators implying a cell-cell crosstalk between infected T cells and MDDCs.

We then addressed the kinetics of the release of these mediators. Conditioned medium of MDDCs cocultured with infected T cells were collected over time, and used to culture fresh autologous MDDCs before the addition of LPS (**Fig 6D**). A lowered responsiveness of MDDCs was only observed with supernatants collected at least 18h after coculture (**Fig 6E**), suggesting that the mediators are produced and released after a transcriptional response. Alternatively, we tested the kinetics required for the mediators to manipulate MDDC responsiveness. Conditioned medium of MDDCs cocultured with infected T cells was collected after 24h, and used to culture fresh autologous MDDCs for a varying duration before the addition of LPS (**Fig 6F**). A lowered responsiveness of MDDCs was only observed when MDDCs were cultured for at least 18h with the conditioned medium before adding the LPS (**Fig 6G**).

Thus, the manipulation of MDDC responsiveness might also rely on a transcriptional control, which is consistent with the unique transcriptional signature induced by exposure to HTLV-1-infected T cells observed by RNA-seq.

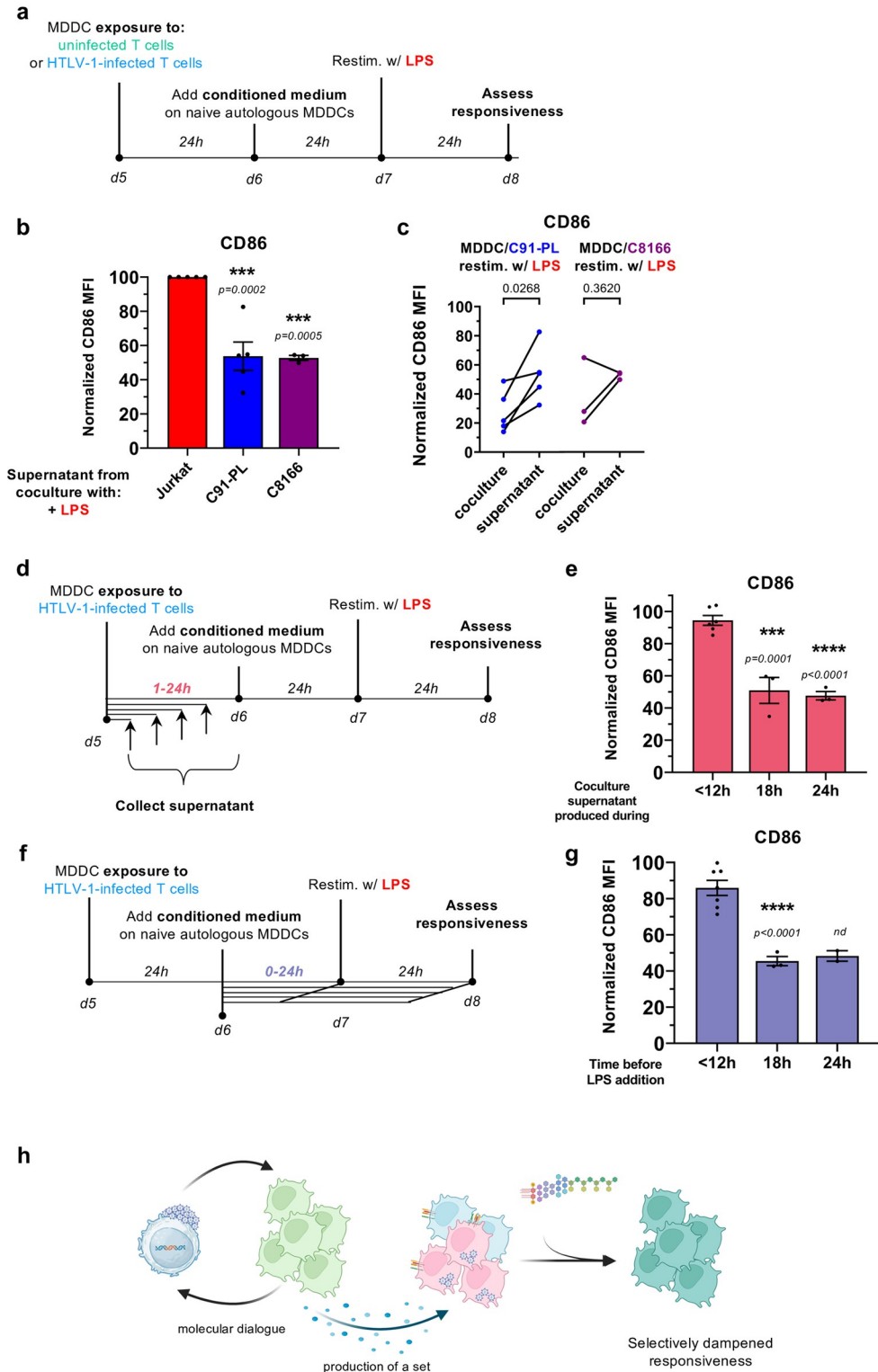

**Fig 6. A cross-talk between HTLV-1-infected T cells and MDDCs induces the release of soluble tolerogenic mediators. a.** Schematic representation of the experimental design. MDDCs were cocultured for 24h with uninfected Jurkat cells, or with HTLV-1-infected C91-PL or C8166 cells. Conditioned medium was collected and added on naive autologous MDDCs for 24h, before restimulation with LPS for an additional 24h. **b.** Flow cytometry analysis after CD11c and CD86 staining. Data are represented as the normalized MFI of CD86, with the MFI in re-stimulated

MDDCs cultured in MDDC/Jurkat-conditioned medium set to 100. The color code is the same as in Fig 5. Data summarize n = 5 (C91-PL) or n = 3 (C8166) independent experiments analysed with ordinary one-way ANOVA as described in S7 Table. **c.** Analysis of paired experiments, after standard coculture with C91-PL (**left**) or C8166 cells (**right**, as measured in **Fig 5A**), or after culture in conditioned medium from MDDC/C91-PL coculture or from MDDC/C8166 coculture (n = 5 or 3 independent experiments, respectively). Data were analysed with Paired t-test as described in S7 Table. **d.** Schematic representation of the experimental setting. MDDCs were cocultured with C91-PL, and conditioned medium was collected at multiple time points of coculture and added on naïve autologous MDDCs for 24h, before re-stimulation with LPS for another 24h. **e.** Flow cytometry analysis after CD11c and CD86 staining. Data are represented as the normalized MFI of CD86, with the MFI in LPS-stimulated MDDCs cultured in normal medium set to 100 (n = 6 and n = 3 independent experiments, for <12h and for 18h and 24h conditions, respectively). Data were analysed with ordinary one-way ANOVA as described in S7 Table. **f.** Schematic representation of the experimental setting. MDDCs were cocultured for 24h with C91-PL. Conditioned medium was collected and added on naive autologous MDDCs for the indicated time, before re-stimulation with LPS for another 24h. **g.** Flow cytometry analysis after CD11c and CD86 staining. Data are represented as the normalized MFI of CD86, with the MFI in LPS-stimulated MDDCs cultured in normal medium set to 100 (independent experiments were performed using n = 7 for <12h condition, n = 3 for 18h condition and n = 2 for 24h condition). Statistical analysis of the two first columns was performed with Unpaired Welch's t-test. **h.** Schematic drawing summarizing the results from Figs 6 and S9. The drawing was created using BioRender.com.

Altogether, our results (recapitulated in **Fig 6H**) show that upon coculture, a reciprocal cross-talk is initiated between HTLV-1-transformed T cells and MDDCs, which results in the transcriptionally-controlled release of a set of soluble tolerogenic mediators that manipulates MDDC responsiveness, in cooperation with additional mechanisms dependent on viral capture and/or cell-cell contacts.

## Discussion

The remarkable ability of viruses to manipulate intracellular pathways in infected host cells is essential for establishing and maintaining infection. In addition to these direct effects, viruses may also establish a specific microenvironment around the infected cells, which may favor viral spread and persistence by influencing the signaling pathways and behavior of neighboring, uninfected cells, including immune cells [28,29]. As a by-stander effect, this may alter immune responsiveness to other stimuli. Mechanisms by which viruses influence uninfected cells in their local environment are still poorly understood. In this study, we aimed to decipher how HTLV-1-infected cells indirectly manipulate neighboring DCs. We demonstrate that exposure to HTLV-1-infected T cells induces a unique transcriptional signature in DCs, preventing their maturation and impairing their ability to be subsequently activated. This induction of tolerogenicity is not directly caused by infection of DCs, but instead results from a cross-talk between infected T cells and DCs, reminiscent of a viral microenvironment [28,29].

Previous work on the interplay between HTLV-1 and DCs has focused on the role of DCs as intermediate target cells for HTLV-1 transmission to T cells [10–12]. Several studies investigated whether viral capture by DCs could lead to T cell cis- and/or trans-infection, and whether specific DC subtypes or maturation statuses could impact this proviral function. In particular, in a previous paper, we demonstrated that immature MDDCs do efficiently capture and transmit HTLV-1 to T cells, while mature MDDCs do not efficiently transmit HTLV-1 to T cells, despite high levels of viral capture [11]. While these data indicated that the maturation status of DCs could impact their proviral functions, they did not address whether exposure to HTLV-1 induced an antiviral immune response by DCs. Such a question was investigated in pDCs, which we showed to be efficient at sensing HTLV-1-infected cells after cell-cell contact, and at responding by producing IFN-I [21]. Here, we specifically addressed how MDDCs respond to HTLV-1 exposure. Intriguingly, our results are in stark contrast to those obtained in pDCs, as we show that MDDCs do not respond to HTLV-1-infected T cells by activating a typical maturation and antiviral program, even in the presence of cell-cell contact. However, a

unique transcriptional response is observed in exposed MDDCs, which could be the result of a specific cross-talk occurring between HTLV-1-infected T cells and MDDCs, that does not occur with pDCs (a graphical comparison of the behavior of HTLV-1-exposed pDCs and MDDCs is depicted in **S10 Fig**). Alternatively, this cross-talk could also occur with pDCs, but be outperformed by the highly efficient capacity of pDCs to produce IFN-I. Investigating how pDC added to the coculture between HTLV-1-infected T cells and MDDCs could modulate this induction of tolerogenicity, and how this tolerogenic behavior would shape HTLV-1 transmission to uninfected T cells, would give insight into how these distinct behaviors are integrated into the microenvironment system. Of note, our results are also in contrast with data obtained on murine DCs, using cell-free Moloney-pseudotyped chimeric HTLV-1[30], which however may not fully recapitulate *in vivo* virus / host interactions in Human. Specifically, given that cell-free HTLV-1 is not detected in HTLV-1 carriers or patients, human DCs are probably mostly exposed *in vivo* to chronically infected cells rather than cell-free virus. This underlines the importance of working with human cells when investigating HTLV-1/host interactions.

Our data point towards the release of tolerogenic mediators following the cross-talk between infected T cells and MDDCs upon coculture, which could cooperate with other mechanisms dependent on viral capture and/or cell-cell contacts. The identity of the released mediators initiating cross-talk observed between HTLV-1-infected T cells and MDDCs, as well as produced as a result of this cross-talk, is currently under investigation in our laboratory. As we observed that medium from infected T cells alone did not induce the tolerogenic behavior of MDDCs, but that conditioned medium from infected T cells cocultured with MDDCs did, it raises the hypothesis that HTLV-1-infected T cells sense a factor released by MDDCs, that activates the release of a second factor by infected T cells. This second factor could then launch a specific transcriptomic response in MDDCs that would lead to tolerogenicity. Among potential mediators released by HTLV-1-infected cells, exosomes emerge as compelling candidates. Their composition could be modified by viral infection, explaining the differences observed in MDDC responses between exposure to uninfected or infected T cells. Also, their production by T cells could be boosted by coculture with MDDCs, explaining the requirement of a cross-talk between both cell types. Last, the diameter of exosomes, around 100 nm, is compatible with their diffusion through the pores of the transwell (0.4 μm) used in our experimental settings. Interestingly, several HTLV-1-infected T cell lines, independently of their ability to produce viral particles or not, do produce exosomes containing cytokines, viral proteins and RNA [31–33] able to modulate target cells, resulting in an increased susceptibility to infection [24]. We can thus hypothesize that in our experimental settings, these exosomes could modulate DC functions, inducing the unique transcriptional signature that we observe. This would be reminiscent of data obtained on Epstein-Barr Virus (EBV), where infection changes the ability of exosomes purified from gastric cancer cell lines to mature MDDCs, leading to a defect in CD86 upregulation and reduced tumor immunity [34].

Whether HTLV-1 proteins contained in exosomes could contribute to the induction of tolerogenicity remains to be investigated. Viral proteins such as p30, HBZ, and Tax are found in exosomes produced by infected cells [24,31], independently of the production of viral particles by these infected cells. In addition, these viral proteins have the potential to interfere with immune functions. For instance, p30 expressed in MDDCs reduces IFN-I responses upon TLR3/4 stimulation, but not upon TLR7/8 stimulation [35], suggesting a specific targeting of the TLR3/4 signaling pathway. However, as we found no alteration in the IFN response of LPS restimulated, HTLV-1-pre-exposed MDDCs, a role of p30 can be excluded. In contrast, HBZ is known to inhibit the NF-κB pathway [36] by degrading the transcription factor p65[37]. As we observed the repression of the NF-κB pathway in HTLV-1-pre-exposed MDDCs upon LPS

restimulation, we could suggest that exosome-transferred HBZ could contribute to modulating MDDC functions.

Our transcriptomic analyses of MDDCs show that following the cross-talk with infected T cells, more than 450 genes are differentially expressed compared to exposure to uninfected T cells, indicating that exposure to infected T cells is not transcriptionally silent. However, this transcriptional response is very partial when compared to LPS stimulation. For instance, among the upregulated genes, we retrieved 121 ISGs, which corresponds to only approximately 30% of all known ISG that respond to LPS stimulation. The amplitude of upregulation of these ISG was also lower than that induced by LPS stimulation, confirming that the induction of ISG in MDDCs in this context is poor. Such a poor induction of ISG might not be sufficient to confer functional responsiveness to MDDCs.

Interestingly, following the cross-talk between infected T cells and MDDCs, we also specifically observed the upregulation of genes involved in fatty acid metabolism (such as *ELOVL3*, *FADS1*, and *SLC27A6*) in MDDCs. As these genes are not found in the typical maturation program of MDDCs, it raises the possibility that upon coculture with HTLV-1-infected T cells, MDDCs produce fatty acids, that may act in an autocrine and paracrine manner to modulate MDDC functions. Of note, fatty acid precursors such as squalene and vitamin D are recognized inducers of tolerogenic DCs [38,39]. Polyinsaturated fatty acids have also been shown to block DC activation and functions [40], and inhibition of fatty acid synthesis has been shown to increase their production of pro-inflammatory cytokines [41]. Whether HTLV-1-exposed MDDCs produce higher levels of fatty acids, and whether these could contribute to their observed tolerogenic behavior, in combination with the poor induction of ISG, is currently under investigation. Thus, our transcriptomic analysis hint at how viruses could indirectly rewire lipid metabolism in immune cells to modulate the immune microenvironment.

Upon restimulation with LPS, HTLV-1-pre-exposed MDDCs undergo a specific transcriptomic response, including an abolished downregulation of *TGFB* expression, which remains highly expressed, and upregulation of *IL13* expression, which could be the basis for their tolerogenic properties. Of note, from the literature, the exact transcriptomic program underlying the tolerogenic behavior of DCs is unclear, as different transcriptomic signatures have been reported [42–44]. Nonetheless, a set of common genes has been defined as a gene signature of tolerogenic DCs [23], including *DDIT4* (DNA-damage-inducible transcript 4), which encodes an inhibitor of mTOR signaling [23]. Interestingly, responsiveness of *DDIT4* was also specifically conferred by pre-exposure to infected cells. Thus, the transcriptomic program induced in HTLV-1-pre-exposed MDDCs by LPS restimulation partially mirrors the tolerogenic DC signature defined in other contexts.

Altogether, our study demonstrates that exposure to HTLV-1-transformed cells can indirectly distort the functionality of DCs and impair their response to subsequent stimulation. Our *in vitro* results are consistent with *in vivo* observations, which report that MDDCs from HTLV-1-infected patients exhibit deficiencies in both basal maturation and responsiveness to TNF-α treatment, as well as defects in the induction of T-cell response [15,17]. In addition, PBMCs from HTLV-1-infected patients stimulated with tuberculin produce fewer TNF-α than PBMC isolated from healthy individuals [45]. Further, to articulate our findings with the mechanisms of HTLV-1 pathogenesis, our transcriptomic results obtained on naïve MDDCs acutely exposed to HTLV-1 may also be confronted to transcriptomic signatures described by others on total blood, PBMCs, or purified CD4+ or CD8+ T cells, from infected individuals, or from T cell lines [46–51] (**S1 Table**). For instance, Tattermusch *et al.* have reported a specific interferon-inducible signature in whole blood from TSP/HAM patients [46]. Among the ISG that we identified as upregulated in MDDCs after exposure to HTLV-1-infected cells, 25 were also detected in the whole blood of asymptomatic carriers, and 9 in the specific signature of

TSP/HAM. This overlap in DEG between studies might remain partial due to the very different experimental setup (MDDCs *vs.* whole blood, acute *in vitro* exposure *vs.* chronically infected individuals). In addition to this interferon-inducible signature, a IL-17 signature has also been associated with the progression to TSP/HAM in HTLV-1-infected individuals [52,53]. While we observed that pre-exposure to HTLV-1 dampens the ability of naïve MDDCs to induce the IL-17 pathway upon stimulation by a TLR agonist, we did detect the upregulation of several genes of the IL-17 pathway in MDDCs upon exposure to HTLV-1-infected cells. This might be consistent with a role of the IL-17 pathway in HTLV-1 pathogenesis, but the biological interpretation of such comparisons should be taken with great caution.

Finally, our results showing that viral production is not necessary to dampen MDDC responsiveness, might reflect the *in vivo* situation in which HTLV-1 expression is repressed in chronically infected cells [54]. Therefore, it is plausible that defects in the myeloid cell response, particularly in DCs, may represent a mechanism of HTLV-1 pathogenesis *in vivo*. Thus, this work illustrates how HTLV-1 might induce a local immune microenvironment suitable for its own persistence. Such a microenvironment may additionally contribute to bystander immune dysfunctions, in asymptomatic HTLV-1 carriers as well as symptomatic HTLV-1-infected patients.

## Methods

### Cells lines

HTLV-1 infected T-cell lines (C91-PL from Cellosaurus ref CVCL_0197, MT-2 ref CVCL_2631[55], Hut102 ref CVCL_3526[56] and C8166 ref ECACC 88051601[22]) and control uninfected T-cell lines (Jurkat from ATCC ref ACC 282, CEM/C1 ref CRL-2265, Molt-4 Cellosaurus ref CVCL_0013) were maintained at a cell density of $0.5.10^6$ cells/mL in complete RPMI medium: RPMI1640 GlutaMAX (Gibco; 61870010) supplemented with 10% fetal calf serum (FCS) and penicillin-streptomycin (100 U/mL and 100 µg/mL respectively).

The human fibrosarcoma cell line HL116 stably expressing the firefly luciferase reporter gene under the control of the immediate early IFN-I inducible 6–16 promoter (kindly provided by Dr. S. Pelligrini, Institut Pasteur, France [57]) was maintained under HAT selection (Gibco; 21060017, used at 1X final concentration) in DMEM GlutaMAX pyruvate medium (Gibco; 10569010) supplemented with 10% FCS and penicillin-streptomycin (100 U/mL and 100 µg/mL respectively).

All cells were grown at 37°C in 5% $CO_2$ and tested negative for mycoplasma contamination on a regular basis. None of the cell lines were authenticated.

### Human primary monocyte-derived dendritic cells (MDDCs)

MDDCs were derived from purified monocytes from healthy blood donors as described previously [11]. Briefly, blood samples were collected at Etablissement Français du Sang (EFS) from anonymous healthy blood donors according to the institutional Standard Operating Procedures for blood donation, including a signed informed consent. Blood was diluted in PBS 1X (Gibco) and peripheral blood mononuclear PBMCs were isolated using a density gradient separation using Ficoll-Paque (Fisher Scientific, 11778538). Monocytes were then isolated from PBMCs using a density gradient separation using Percoll Centrifugation Medium (Fisher Scientific, 10607095). Freshly or frozen monocytes were cultured in six-well plates at $3.10^6$ cells/mL in complete MDDC medium: RPMI1640 GlutaMAX (Gibco) supplemented with 10% FCS, penicillin-streptomycin (100 U/mL and 100 µg/mL respectively), Hepes buffer (Gibco, 15630080; 10 mM), MEM non-essential amino acids (2,5mM, Gibco, 11140050), sodium

pyruvate (Gibco, 11360070; 1 mM), and beta-mercaptoethanol (Gibco, 31350–010; 0.05 mM). MDDC medium was supplemented with IL-4 and GM-CSF (Miltenyi Biotec, 130-093-922 and 130-093-866; 100 ng/mL each) for differentiation. On day 3, the culture medium was refreshed by discarding half of the medium and adding the same volume of new MDDC medium and twice concentrated IL-4 and GM-CSF to all cell cultures. Immature MDDCs were harvested on day 5 or 6. Every experiment was repeated on MDDCs derived from independent blood donors.

### Reagents and antibodies

Toll-like receptor (TLR)-4 agonist (LPS, tlrl-3pelps; 1μg/mL) and TLR-7/8 agonist (R848, tlrl-r848; 3μg/mL) were purchased from Invivogen. MeV IC323-eGFP [58] is a recombinant MeV expressing the gene-encoding eGFP (using the plasmid-encoding MeV IC323-eGFP kindly provided by Yanagi, Kyushu University, Fukuoka, Japan). MeV IC323 recombinant virus was rescued in 293-3-46 cells, as previously described [59]. Production of the viruses was performed at 32˚C. All viruses were propagated and titrated in Vero-SLAM/CD150 cells. MDDCs were infected using a multiplicity of infection (MOI) of 1.

The following antibodies were used: V450-coupled mouse anti-Human CD11c (BD Biosciences, 560369; 1/100), PE-coupled mouse anti-Human CD86 (Invitrogen, 12-0869-42; 1/100), APC-coupled mouse anti-Human CD83 (Miltenyi Biotec, 130-094-186; 1/50), BV510-coupled mouse anti-Human CD83 (BD Biosciences, 563223; 1/50), APC-H7-coupled mouse anti-Human CD80 (BD Biosciences, 561134; 1/50), BB515-coupled mouse anti-Human PD-L1 (BD Biosciences, 564554; 1/25), PE-Cy7-coupled mouse anti-Human ICOSL (BioLegend, 309410; 1/100), Mouse anti-Gagp19 clone TP7 (Zeptometrix, 081107; 1/500), AlexaFluor647-coupled Alpaca anti-Mouse IgG1 (Chromotek, sms1AF647-1-5; 1/500), Biotin-coupled mouse anti-Human CADM1 (MBL Life Sciences, CM004-6; 1/1000), AlexaFluor647-coupled mouse streptavidin (Invitrogen, S31374; 1/500).

### MDDCs coculture experiment

Immature MDDCs ($3.10^5$ cells) were plated in 48-well plates and cocultured with HTLV-1-infected or control uninfected T-cells ($6.10^4$ cells) in 300μL complete MDDC medium supplemented with IL-4 and GM-CSF for 24h or 48h. When indicated, transwell 24-well plates with permeable polycarbonate membrane inserts (0.4μm diameter) were used (Fisher Scientific, 10147291). Alternatively, MDDCs were cultured in supernatant collected from either HTLV-1-infected or control uninfected T-cell culture, or from coculture of MDDCs with HTLV-1-infected cells or control uninfected T-cell during 24h (or the indicated duration). After 24h, cells were harvested, or stimulated with LPS or R848 or medium for an additional 24h (or the indicated duration). At the indicated time points, cells and supernatants were collected for phenotyping using flow cytometry and for cytokine quantification, respectively.

### Flow cytometry analysis

To assess surface marker expression, cells were fixed using 4% paraformaldehyde (PFA) diluted from 20% PFA (Electron Microscopy Science, 50-980-493) in PBS 1X, and stained with the indicated surface markers antibodies diluted in PBS 1X-1% Bovine Serum Albumin (BSA) for 20 min at 4˚C. To assess viral capture, cells were fixed and permeabilized using the Fix/Perm FoxP3 and Transcription factors kit (Invitrogen, 00-5523-00) according to the manufacturer's instructions, and stained with mouse anti-Gagp19 antibody followed by the AlexaFluor647-coupled anti-mouse antibody diluted in the PERM buffer supplemented with 7% Normal Goat Serum (NGS) for 25 min at room temperature. Cells were then stained for the

indicated surface markers diluted in PBS 1X 1% BSA as described above. Compensation beads (BD Biosciences, 552843) were used to correct signal overlap between the emission spectra of the different fluorophores.

Data were acquired using a flow cytometer FACS CantoII (BD Biosciences) and analyzed with FlowJo v10.7 software (BD Life Sciences). The full gating strategy is exemplified in **S1 Fig**.

### Cytokine quantification

TNF-α and IL-10 were quantified in the collected supernatants using Bio-Plex Pro Human Luminex kit (Bio-Rad) according to the manufacturer's instructions.

### Type I interferon quantification

HL116 cells were seeded at $2.10^4$ cells/well in 96-U-bottom-well plates and incubated for 24h. Supernatant collected from MDDCs culture (100 μL) or serial dilutions of recombinant IFN-α (Tebu-Bio, RPA033Gu02) used for standard curve determination were added for an additional 17h. Cells were then lysed (Promega Passive Lysis Buffer, E1941) and luciferase activity assayed according to the manufacturer's instruction (Promega Luciferase Assay System, E1501).

### MDDCs isolation after coculture

Cells from MDDC/T-cell lines coculture ($1.10^6$ MDDCs and $2.10^5$ T-cells) were harvested and stained using biotin-coupled anti-CADM1 followed by AlexaFluor647-coupled streptavidin diluted in PBS 1X-1% BSA for 20 min at 4°C. After several washes in PBS, cells were then incubated with anti-AlexaFluor647 MicroBeads (Miltenyi Biotec, 130-091-395) and separated on MACS Separation LD Columns (Miltenyi Biotec, 130-042-901) according to the manufacturer's instructions. To assess the purity and yield of enrichment of MDDCs after magnetic separation, $6.10^4$ cells were collected before and after separation on LD columns, fixed in 4% PFA and stained with CD11c-V450 antibody, before analysis using FACS Canto II.

### RNA preparation

After magnetic separation, MDDCs were lysed and total RNA was extracted using the NucleoSpin RNA Mini kit for RNA purification (Macherey-Nagel, 740955) according to the manufacturer's instructions. RNA concentration was determined with a NanoDropND1000 spectrophotometer (Thermo Fisher Scientific) and samples were stored at -80°C until shipment for external sequencing. Stranded RNA libraries were prepared after removal of rRNA. High throughput sequencing of 150 bp paired-end reads was performed with an Illumina HiSeq 2500 platform by Novogene Europe (Cambridge, United Kingdom). Each sample had on average 50 million matched pairs of reads.

### RNA-seq data analysis

The quality of sequences was checked using the FastQC tool. The reads were trimmed with PrinSeq [60] to remove low-quality bases and then mapped to the human reference transcriptome (hg19) using Kallisto pseudoalignment [61]. Differential gene analysis was carried out with DESeq2 package [62]. Genes with a basemean >10 showing an absolute fold change (FC) ≥ 2 with an adjusted p-value <0.05 (Wald test using Benjamini and Hochberg method) were considered differentially expressed. All DEGs identified in this study are listed in S1 and S4 Tables. DAVID functional annotation tool using the KEGG pathways database was used for gene ontology analysis [63,64]. For clarity, KEGG pathways were filtered and grouped as listed in S2 Table to reduce redundancy.

To identify the expression pattern responsiveness in LPS-stimulated MDDC pre-exposed to C91-PL, we retrieved the genes by Venn analysis using the following list of DEGs: MDDC/Jurkat restim.w/LPS vs MDDC/Jurkat; MDDC/C91-PL restim.w/LPS vs MDDC/Jurkat restim.w/LPS; MDDC/C91-PL restim.w/LPS vs MDDC/Jurkat. Then, genes were classified as follows (summarized in **S7A Fig**): (i) responsiveness confered by pre-exposure to infected cells: genes that are not differentially expressed between mock- and LPS-stimulated Jurkat-pre-exposed MDDCs, but are differentially expressed (either up or down) between mock-stimulated Jurkat-pre-exposed MDDCs and LPS-stimulated C91-PL-pre-exposed MDDCs; (ii) responsiveness exacerbated by pre-exposure to infected cells: genes that are either up-regulated in LPS-stimulated Jurkat-pre-exposed MDDCs, compared to mock-stimulated Jurkat-pre-exposed MDDCs, and further up-regulated in LPS-stimulated C91-PL-pre-exposed MDDCs, compared to LPS-stimulated Jurkat-pre-exposed MDDCs; or down-regulated in LPS-stimulated Jurkat-pre-exposed MDDCs compared, to mock-stimulated Jurkat-pre-exposed MDDCs, and further down-regulated in LPS-stimulated C91-PL-pre-exposed MDDCs, compared to LPS-stimulated Jurkat-pre-exposed MDDCs; (iii) responsiveness attenuated by pre-exposure to infected cells: genes that are either up-regulated in LPS-stimulated Jurkat-pre-exposed MDDCs, compared to mock-stimulated Jurkat-pre-exposed MDDCs, as well as up-regulated in LPS-stimulated C91-PL-pre-exposed MDDCs, compared to mock-stimulated Jurkat-pre-exposed MDDCs, but down-regulated in LPS-stimulated C91-PL-pre-exposed MDDCs, compared to LPS-stimulated Jurkat-pre-exposed MDDCs; or down-regulated in LPS-stimulated Jurkat-pre-exposed MDDCs, compared to mock-stimulated Jurkat-pre-exposed MDDCs, in LPS-stimulated C91-PL-pre-exposed MDDCs, compared to mock-stimulated Jurkat-pre-exposed MDDCs, but up-regulated in LPS-stimulated C91-PL-pre-exposed MDDCs, compared to LPS-stimulated Jurkat-pre-exposed MDDCs; (iv) responsiveness abolished by pre-exposure to infected cells: genes that are either up-regulated in LPS-stimulated Jurkat-pre-exposed MDDCs, compared to mock-stimulated Jurkat-pre-exposed MDDCs, as well as down-regulated in LPS-stimulated C91-PL-pre-exposed MDDCs, compared to LPS-stimulated Jurkat-pre-exposed MDDCs, but not differentially expressed in LPS-stimulated C91-PL-pre-exposed MDDCs, compared to mock-stimulated Jurkat-pre-exposed MDDCs; or down-regulated in LPS-stimulated Jurkat-pre-exposed MDDCs compared to mock-stimulated Jurkat-pre-exposed MDDCs, as well as up-regulated in LPS-stimulated C91-PL-pre-exposed MDDCs, compared to LPS-stimulated Jurkat-pre-exposed MDDCs, but not differentially expressed in LPS-stimulated C91-PL-pre-exposed MDDCs, compared to mock-stimulated Jurkat-pre-exposed MDDCs.

## Statistical analysis

Median fluorescence intensity and percentages of gated cells were determined from flow cytometry analysis with FlowJo software. Data were analyzed using Prism 8 (GraphPad Software). The central tendency represents the mean, and the error bar represents the standard error of the mean (SEM). The detailed statistical analysis and associated values are presented in S7 (relative to Figs 1–6) and S8 Tables (relative to S1–S10 Figs). Statistical analysis was performed as follows: normality of the dataset was tested using Shapiro Wilk test. If the dataset followed a Gaussian distribution, differences among means were tested using Repeated measures one-way ANOVA, or Ordinary one-way ANOVA when the dataset contained missing values. Post-hoc comparisons with the mean of the control condition (as defined below) were assessed with Dunnet's test. Note that for Fig 1E, Sidak's test was used to compare the means of the C91-PL-exposed MDDCs depending on their capture status.

When normality of the dataset was not met, analysis was performed using Friedman test, or Kruskal Wallis test when the dataset had missing values. Post-hoc comparisons with the mean of the control condition (as defined below) were performed using Dunn's test.

Multiple comparisons were computed considering MDDCs exposed to uninfected Jurkat T cells as a control condition (p-values reported on the Figures). When relevant, additional multiple comparisons considering MDDCs exposed to HTLV-1-infected C91-PL T cells as a control condition were added (p-values reported in **S7 Table**).

Data presented in **S2 Fig** were also analysed using nested one-way ANOVA with Tukey's multiple comparisons test, grouping cell lines according to the infection status (MDDCs exposed to non-infected cells: Jurkat, CEM, Molt-4; MDDCs exposed to HTLV-1-infected cells: C91-PL, MT-2, Hut102; MDDCs stimulated with LPS), and the detailed valued are presented in **S8 Table**.

For main figures that are subsets of larger supplementary figures, the indicated p-values are reported from the analysis on the total dataset.

When only two groups were compared, statistical analysis was performed using parametric paired t-test (Figs 5C, 6C and S8B) or its non-parametric equivalent Wilcoxon matched-pairs signed rank test (Fig 5C). Note that a parametric unpaired Welch's t-test was used to analyze Fig 6G due to the uneven number of replicates.

The potential correlation investigated in S2I Fig was fitted with a simple linear regression model.

p-value $< 0.05$ was considered significant, and statistics are denoted as * $p<0.05$, ** $p<0.01$, *** $p<0.001$, **** $p<0.0001$.

## Supporting information

**S1 Fig. Gating strategy used in this study.** Example of the hierarchical gating strategy used to analyze MDDC phenotype. Control MDDCs (top panel), MDDCs treated with LPS for 24h (middle panel) or MDDCs cocultured with HTLV-1-infected C91-PL T cells (bottom panel) were identified based on their expression of CD11c, which is absent on T cells, and their maturation status was determined by their expression of CD86 (or other maturation or inhibition markers, see S2A–S2F Fig).
(TIFF)

**S2 Fig. Human MDDCs do not fully mature when exposed to HTLV-1-infected T cells.** This figure is relative to Fig 1**A**-1**F**. MDDCs were cocultured with control uninfected (Jurkat, CEM, Molt-4, green bars), or HTLV-1-infected T cell lines (C91-PL, MT-2, Hut102, blue bars) for 24h or 48h, as indicated. As a control, MDDCs were stimulated with LPS for 24h ("24h" condition), or left untreated for 24h followed by 24h of LPS treatment ("48h" condition, red bars). Flow cytometry analysis after CD11c and CD86 (**a**), CD83 (**b**), CD80 (**c**), CD40 (**d**), ICOSL (**e**) and PD-L1 (**f**) staining. Data are represented as percentage of positive MDDCs among total CD11c$^+$ MDDCs (left), or as normalized MFI (right) for n = 3–31 independent experiments. Detailed statistical analysis is presented in S8 Table. Briefly, data were analysed with ordinary one-way ANOVA or Kruskal-Wallis test, in accordance with the distribution of the dataset. A complementary analysis with cell lines grouped according to infection status (MDDC exposed to non-infected cells: Jurkat, CEM, Molt-4; MDDC exposed to HTLV-1-infected cells: C91-PL, MT-2, Hut102; MDDC stimulated with LPS) was performed with nested one-way ANOVA, as presented in S8 Table. **g.** Supernatant from the indicated cocultures or from LPS-stimulated MDDCs was collected, and TNF-α (left) and IFN-I (right) were quantified by Luminex and reporter cell assay, respectively. Results from n = 3 or 8–5 independent experiments, respectively. Data were analysed with RM one-way ANOVA or Kruskal-

Wallis test, respectively, and with nested one-way ANOVA, grouping cell lines according to the infection status, as presented in S8 Table. **h.** Viral capture in MDDCs cocultured with Jurkat, C91-PL, MT-2 or Hut102 cells was assessed by Gag p19 staining on CD11c$^+$ MDDCs. The percentage of Gag p19$^+$ MDDCs was determined in n = 7–14 independent experiments. Data were analysed with Kruskal-Wallis test, as presented in S8 Table. **i.** The percentage of CD86$^+$ MDDCs from repeated experiments was plotted against the percentage of Gag p19$^+$ MDDCs (n = 15), and a linear correlation fit curve was added, according to the simple linear regression analysis performed, as presented in S8 Table.
(TIFF)

**S3 Fig. Purity and yield of MDDCs isolation after negative magnetic selection based on CADM1 expression. a.** CADM1 expression on MDDCs, Jurkat or C91-PL cells. Expression levels in individual cell types were superimposed (right) to show the exclusive expression of CADM1 on T cells compared to MDDCs, with higher expression levels on C91-PL cells compared to Jurkat cells. **b.** MDDC were cocultured for 24h with Jurkat (left) or C91-PL T cells (right), and cells from each coculture condition were stained for CD11c before or after negative magnetic selection based on CADM1. Representative flow cytometry plots are shown. The percentages of T cells (Jurkat, green, or C91-PL, cyan) and of MDDCs (magenta), before or after isolation of MDDCs, are indicated.
(TIFF)

**S4 Fig. Exposure to HTLV-1-infected T cells induces a unique transcriptional signature in MDDCs.** This figure is relative to Fig 2. **a.** Heatmap of the 474 total DEGs in C91-PL-exposed MDDCs compared to Jurkat-exposed MDDCs. **b.** DESeq2 normalized counts for *CCL19*, *TNFSF13B*, *TRIM25*, *BCL2L1* and *BCL2A1* across samples. **c.** DESeq2 normalized counts for *RELA*, *RELB* and *NFKB1* across samples. **d.** Heatmap of *ISG*s expression across samples.
(TIFF)

**S5 Fig. Pre-exposure to HTLV-1-infected T cells dampens the responsiveness of MDDCs to subsequent stimulation.** This figure is relative to Fig 3. MDDCs were cocultured with uninfected (Jurkat, CEM, Molt-4, red bars) or HTLV-1-infected T cells (C91-PL, MT-2, Hut102, dark blue bars) for 24h, before restimulation with LPS (a-c, e) or R848 (d) for an additional 24h. **a-d.** Flow cytometry analysis after CD11c and CD86, CD83 and CD80 staining. The normalized MFI (left) and percentage of positive MDDCs (right) was determined in n = 5–30 independent experiments. Detailed statistical analysis is presented in S8 Table. Briefly, data were analyzed with ordinary one-way ANOVA or Kruskal-Wallis test, in accordance with the distribution of the dataset. **e.** Supernatant from the indicated cocultures was collected after LPS restimulation, and TNF-α (left) and IFN-I (right) concentrations were quantified for n = 10 or 4–5 independent experiments, respectively. Data were analyzed using RM one-way ANOVA or ordinary one-way ANOVA, respectively, as presented in S8 Table.
(TIFF)

**S6 Fig. Viral capture in MDDCs cocultured with C91-PL or C8166 T cells.** Viral capture in MDDCs cocultured with C91-PL or C8166 T cells was assessed after 24h of coculture by Gag p19 staining on CD11c$^+$ MDDCs. The percentage of Gag p19$^+$ MDDCs among total CD11c$^+$ MDDCs was determined. Left: representative experiment. Right: Results from n = 14 independent experiments. Data were analysed with Friedman test, as presented in S8 Table.
(TIFF)

**S7 Fig. Pre-exposure to HTLV-1-infected T cells alters the transcriptional response of MDDCs to subsequent stimulation.** This figure is relative to Fig 4. **a.** Recapitulative table

summarizing the different gene responsiveness patterns observed in the dataset (see Material and Methods for details). Genes were classified by comparing the DEG lists retrieved from the 3 comparisons indicated on the top of the table (criteria). A theoretical expression pattern across samples is illustrated in the right column. **b.** DESeq2 normalized counts across samples of selected genes showing an "attenuated or abolished" (top), "conferred" (middle) or "unaffected" (bottom) responsiveness pattern.
(TIFF)

**S8 Fig. Neither HTLV-1 viral capture, nor cell-cell contact with infected T cells, are strictly required to dampen the responsiveness of MDDCs to subsequent stimulation.** This figure is relative to Fig 5. **a.** Flow cytometry analysis after staining for CD11c and CD83 (top), CD11c and CD80 (middle), or CD11c and C86 (bottom). Data are represented as percentage of CD83+ (CD80+ or CD86+ respectively) MDDCs (left), or as normalized MFI of CD83 (CD80 or CD86 respectively) (right), with the MFI in re-stimulated Jurkat-pre-exposed MDDCs set to 100, and summarize n = 5 independent experiments. Red: Jurkat-pre-exposed MDDC restimulated with LPS or R848; dark blue: C91-PL pre-exposed MDDC restimulated with LPS or R848; purple: C8166 pre-exposed MDDC restimulated with LPS or R848. Detailed statistical analysis is presented in S8 Table. Briefly, data were analysed with RM one-way ANOVA Friedman test, in accordance with the distribution of the dataset. **b.** Representative cytometry plots (left) showing the viral capture in MDDCs cocultured with C91-PL T cells, in the absence (standard coculture) or presence (transwell) of a transwell insert (0.4μm pore diameter), as assessed by Gag p19 staining on CD11c+ MDDCs. The quantification of data from n = 5 independent experiments are summarized (right). Data were analysed with a paired t-test, as presented in S8 Table.
(TIFF)

**S9 Fig. Conditioned medium from HTLV-1-infected T cells is not sufficient to dampen the responsiveness of MDDCs to subsequent stimulation. a**. Quantification by Luminex of IL-10 levels in supernatants of MDDCs cocultured with uninfected T cells (Jurkat), or with HTLV-1-infected T cells (C91-PL, MT-2, Hut102, C8166), followed or not by LPS stimulation (n = 1 experiment). **b.** Uninfected (Jurkat, red) or HTLV-1-infected T cells (C91-PL, MT-2, Hut102, dark blue; or C8166, purple) were cultured for 24h. Conditioned medium was collected and added on naive MDDCs for 24h, before restimulation with LPS for an additional 24h. Quantification results from n = 4 independent experiments of flow cytometry data for CD11c and CD86 staining are presented. Data were analysed with RM one-way ANOVA, as presented in S8 Table.
(TIFF)

**S10 Fig. Graphical comparison of the behavior of HTLV-1-exposed MDDCs and pDCs. a.** Summary of the results presented in this study: a molecular dialogue established between MDDCs and HTLV-1-transformed T-cells upon coculture, partially dependent on viral capture and cell-cell contact, leads to the release of soluble mediators that impair MDDCs maturation and dampen their responsiveness. **b.** Summary of pDCs response to exposure of HTLV-1-infected cells as reported in [21]: pDCs senses HTLV-1 enveloped virions after contact with HTLV-1-infected T-cells mediated by HTLV-1 biofilm. This results in TLR7-dependent pDCs activation and high production of IFN-I, the amount of which is regulated by biofilm composition. The drawing was created using BioRender.com.
(TIFF)

**S1 Table. Lists of DEGs relative to Figs 2 and S4.**
(XLSX)

**S2 Table. Lists of filtered and grouped KEGG pathways relative to Figs 2C and 4C.** Enrichment values and p-values are given for each individual pathway ("Individual KEGG pathway" sheet), and the calculated means for grouped pathways are given in the "Grouped KEGG pathway" sheet.
(XLSX)

**S3 Table. Lists of DEGs in each KEGG pathway, relative to Fig 2C.** All genes annotated in each KEGG pathway are listed, and those retrieved among the DEGs in our dataset are highlighted in grey. For grouped pathways, the "Total" column summarizes all genes from the individual grouped pathways.
(XLSX)

**S4 Table. Lists of DEGs relative to Fig 4.**
(XLSX)

**S5 Table. List of downregulated DEGs in each KEGG pathway, relative to Fig 4C.** All genes annotated in each KEGG pathway are listed, and those retrieved among the downregulated DEGs in our dataset are highlighted in grey. For grouped pathways, the "Total" column summarizes all genes from the individual grouped pathways.
(XLSX)

**S6 Table. List of upregulated DEGs in each KEGG pathway, relative to Fig 4C.** All genes annotated in each KEGG pathway are listed, and those retrieved among the upregulated DEGs in our dataset are highlighted in grey. For grouped pathways, the "Total" column summarizes all genes from the individual grouped pathways.
(XLSX)

**S7 Table. List of Source data relative to figures.** All raw data corresponding to the MFI or % for each gating defined by FlowJo, as well as statistical analyses, tests used and degree of freedom are listed for each individual panel in each figure, with the exception of RNAseq data from Figs 2 and 4.
(XLSX)

**S8 Table. List of Source data relative to S1–S10 Figs.** All raw data corresponding to the MFI or % for each gating defined by FlowJo, as well as statistical analyses, tests used and degree of freedom are listed for each individual panel in each figure, with the exception of RNAseq data from S4 and S7 Figs.
(XLSX)

## Acknowledgments

We dedicate this work to the memory of Pr. Renaud Mahieux. We thank the Retroviral Oncogenesis team, as well as Dr. Arnaud Moris, Dr. Sylvain Baize, Dr. Anne-Sophie Beignon, Dr. Franck Halary, Dr. Claudine Pique, Pr. Mathias Faure, Dr. Delphine Muriaux and Dr. Denis Gerlier for helpful discussion and/or critical reading of the manuscript. This work has benefited from the facilities and expertise of the SFR Biosciences Lyon (UMS3444/CNRS, US8/Inserm, ENS de Lyon, UCBL). We acknowledge the contribution of the Etablissement Français du Sang Auvergne—Rhône-Alpes, and we thank Dr. Marlène Dreux (CIRI) and her team for the help with PBMC isolation. We thank Dr. Carine Rey (BIBS, CIRI, Lyon) for her help with RNA-seq data analysis.

## Author Contributions

**Conceptualization:** Chloé Journo, Hélène Dutartre.

**Data curation:** Auriane Carcone, Chloé Journo, Hélène Dutartre.

**Formal analysis:** Auriane Carcone, Franck Mortreux, Cyrille Mathieu, Chloé Journo, Hélène Dutartre.

**Funding acquisition:** Hélène Dutartre.

**Investigation:** Auriane Carcone, Cyrille Mathieu.

**Methodology:** Auriane Carcone, Sandrine Alais.

**Project administration:** Hélène Dutartre.

**Supervision:** Hélène Dutartre.

**Validation:** Franck Mortreux, Chloé Journo, Hélène Dutartre.

**Writing – original draft:** Auriane Carcone, Hélène Dutartre.

**Writing – review & editing:** Auriane Carcone, Franck Mortreux, Sandrine Alais, Cyrille Mathieu, Chloé Journo, Hélène Dutartre.

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
