## [Decision Letter · Decision Letter 0]

7 Aug 2024

Dear Dr. Dutartre,

Thank you very much for submitting your manuscript "Peculiar transcriptional reprogramming with functional impairment of dendritic cells upon exposure to transformed HTLV-1-infected cells" for consideration at PLOS Pathogens. As with all papers reviewed by the journal, your manuscript was reviewed by members of the editorial board and by three independent reviewers who are experts in the field. The reviewers appreciated the attention to an important topic. Based on the reviews, we are likely to accept this manuscript for publication, providing that you modify the manuscript according to the review recommendations. In particular, reviewer 2 raised several minor concerns that should be addressed.

Sincerely,

Edward William Harhaj, Ph.D.

Academic Editor

PLOS Pathogens

Susan Ross

Section Editor

PLOS Pathogens

Michael Malim

Editor-in-Chief

PLOS Pathogens

orcid.org/0000-0002-7699-2064

Reviewer Comments (if any, and for reference):

Reviewer's Responses to Questions

**Part I - Summary**

Reviewer #1: This article analyses the molecular mechanisms of suppression of MDDC function by HTLV-1 infection. There have been various reports on the role of DCs in HTLV-1 infection. However, the suppression of MDDC function and its molecular mechanisms have not been elucidated yet. In this paper, the authors clearly describe the suppression of MDDC function at the molecular level and clarify the molecular mechanism of functional suppression by HTLV-1-infected cells. Although the identification of the specific molecules involved in the suppression remains a challenge for the future, the paper is commendable as it advances our understanding of the existence and molecular mechanism of the suppression through a meticulous experimental design.

Reviewer #2: In this manuscript, Carcone and coworkers investigate the broad effects of HTLV-1-infected T cells upon human monocyte-derived DC functions, as measured by both flow cytometry and transcriptomic analysis, complemented with functional assays. They show that co-culture with HTLV-1-infected T cells induces a unique transcriptional signature in monocyte-derived DCs, associated with an inefficient maturation and a poor responsiveness to subsequent stimulation LPS. Induction of this functional impairment requires prolonged coculture with HTLV-1-infected cells, and is partially dependent on viral capture, cell-cell contact, and soluble mediators. These data are crucial to enhance our understanding of innate immunity against HTLV-1, and might have clinical implications for HTLV-1-associated pathologies. Major strengths of the study are the use of complimentary technologies and data-driven system biology analysis, and validation by functional assays (co-cultivation, Transwell, viral capture). Limitations are listed below:

Minor:

1. Jurkat is equal to C91-PL in Fig. 1b, this is a bit weird as a starting point since the manuscript focuses on HTLV-1-infected T-cells (although later on strong differences are shown in transcriptomics) Likewise, the statistical analysis of Fig. 1 as detailed in Suppl. Table 7 takes Jurkat as a basis for the multiple comparison correction. While the statistical significance will probably not change, it is important that the statistical tests reflect the research hypothesis, i.e. testing the effect of HTLV-1-infected cells rather than uninfected Jurkat cells. The same goes for other muiltiple-comparison tests in the manuscript.

2. About the role of ISG and type I IFN: the top 100 most upregulated genes in Sup Table 1 are almost exclusively ISG, several of which are present in the HAM disease signature (Tattermusch et al.). As outlined above, I feel the difference between C91-PL and Jurkat is an important finding that merits more discussion. Several of the genes revealed in the transcriptomic analysis across the manuscript FAS/STAT1 and ISGs are found in HAM, especially IL17 signaling has been shown by several groups (Leal et al., Assone et al…..) and confirmed by a recent systematic review (Shafiei et al. Cytokine 2024). Please compare your proposed signature to published HTLV-1 transcriptomic datasets (publicly available) of other groups in either HTLV-1 infected T-cells (Moens et al., Kashima group from Brazil), and particularly in HAM (Tattermusch et al.) and ATL patients (Watanabe group Kataoka et al., Dierckx et al.).

3. A minor follow-up question: can the proposed signature be tolerogenic if ISG and (at least a partial) IFN signature is detected?

Reviewer #3: In this paper A. Carcone et al. titled “Peculiar transcriptional reprogramming with functional impaierment of dendritic cells upon exposure to transformed HTLV-1 infected cells”, tackle a central issue in HTLV-1 pathogenesis, that once understood will have wide implications in the development of immune therapy approaches and preventive vaccines.

This group prior work uncovered the effect of HTLV-1 infection on pDC, whereas the current work investigate the effect of HTLV-1 on human monocytes derived DC (MDDC) , assessing the functionality of MDDC, carrying or not virus.

The work provides in vitro evidence that HTLV-1 causes a protracted functional impairment of MDDC via indirect mechanisms. Soluble factor (s) and cell to cell contact not dependent on expression of structural genes such as the Gag and Envelope proteins, as the results were recapitulated using the non-producer virus infected T-cell line 8166 that carries only completely spliced viral mRNAs due to a Rex functional defect.

The current work does not explore whether in vivo HTLV-1 infection of humans causes also defect in DC maturation and cytokines production. However, given the complexity of the findings reported here, to address this point will require a substantial amount of work on a properly powered cohort of infected and uninfected people willing to donate samples to address the question. Thus, in my opinion the lack of human data should not penalize this paper.

Similarly whether Tax , HBZ or other viral genes cause the DC maturation defect has been only discussed, but again to address this point will require a large amount of work.

In summary the data presented are convincing, complex but well explained, and I do not have technical criticisms or suggestions.

This is an important and novel paper and my only suggestion is to draw a picture representing the effect of HTLV-1 infection on both pDC and mDC to summarize graphically the work on DC performed over the years.

**Part II – Major Issues: Key Experiments Required for Acceptance**

Reviewer #1: To clarify the effects of HTLV-1-infected cells on MDDCs, the authors clearly demonstrated in a careful experimental design, from the confirmation of maturation inhibition by co-culture to the fact that the gene expression regulatory effects of co-culture are brought about by liquid factors, not through cell contact.

Although the identification of soluble factors that exert maturation inhibition is the subject of subsequent experiments, the authors have clearly clarified the reality of differentiation inhibition and gene expression regulation by the interaction between HTLV-1-infected cells and MDDCs.

Thus, the remaining issues are characterization and identification of the released mediators initiating crosstalk between HTLV-1-infected T cells and MDDCs. However, this experiment is not required in the present manuscript.

Reviewer #2: No major issues

Reviewer #3: none

**Part III – Minor Issues: Editorial and Data Presentation Modifications**

Reviewer #1: There appears no need for modifications.

Reviewer #2: Minor:

1. Jurkat is equal to C91-PL in Fig. 1b, this is a bit weird as a starting point since the manuscript focuses on HTLV-1-infected T-cells (although later on strong differences are shown in transcriptomics) Likewise, the statistical analysis of Fig. 1 as detailed in Suppl. Table 7 takes Jurkat as a basis for the multiple comparison correction. While the statistical significance will probably not change, it is important that the statistical tests reflect the research hypothesis, i.e. testing the effect of HTLV-1-infected cells rather than uninfected Jurkat cells. The same goes for other muiltiple-comparison tests in the manuscript.

2. About the role of ISG and type I IFN: the top 100 most upregulated genes in Sup Table 1 are almost exclusively ISG, several of which are present in the HAM disease signature (Tattermusch et al.). As outlined above, I feel the difference between C91-PL and Jurkat is an important finding that merits more discussion. Several of the genes revealed in the transcriptomic analysis across the manuscript FAS/STAT1 and ISGs are found in HAM, especially IL17 signaling has been shown by several groups (Leal et al., Assone et al…..) and confirmed by a recent systematic review (Shafiei et al. Cytokine 2024). Please compare your proposed signature to published HTLV-1 transcriptomic datasets (publicly available) of other groups in either HTLV-1 infected T-cells (Moens et al., Kashima group from Brazil), and particularly in HAM (Tattermusch et al.) and ATL patients (Watanabe group Kataoka et al., Dierckx et al.).

3. A minor follow-up question: can the proposed signature be tolerogenic if ISG and (at least a partial) IFN signature is detected?

Reviewer #3: I suggest a graphical representation own the effect of HTLV-1 on pDC and mDC

PLOS authors have the option to publish the peer review history of their article (what does this mean?). If published, this will include your full peer review and any attached files.

Reviewer #1: **Yes: **Toshiki Watanabe

Reviewer #2: No

Reviewer #3: No

Figure Files:

Data Requirements:

Reproducibility:

References:

---

## [Decision Letter · Decision Letter 1]

30 Aug 2024

Dear Dr. Dutartre,

We are pleased to inform you that your manuscript 'Peculiar transcriptional reprogramming with functional impairment of dendritic cells upon exposure to transformed HTLV-1-infected cells' has been provisionally accepted for publication in PLOS Pathogens.

Best regards,

Edward William Harhaj, Ph.D.

Academic Editor

PLOS Pathogens

Susan Ross

Section Editor

PLOS Pathogens

Michael Malim

Editor-in-Chief

PLOS Pathogens

orcid.org/0000-0002-7699-2064

Reviewer Comments (if any, and for reference):

Reviewer's Responses to Questions

**Part I - Summary**

Reviewer #2: (No Response)

**Part II – Major Issues: Key Experiments Required for Acceptance**

Reviewer #2: (No Response)

**Part III – Minor Issues: Editorial and Data Presentation Modifications**

Reviewer #2: (No Response)

PLOS authors have the option to publish the peer review history of their article (what does this mean?). If published, this will include your full peer review and any attached files.

Reviewer #2: No

---

## [Editor Report · Acceptance letter]

10 Sep 2024

Dear Dr Dutartre,

We are delighted to inform you that your manuscript, "Peculiar transcriptional reprogramming with functional impairment of dendritic cells upon exposure to transformed HTLV-1-infected cells," has been formally accepted for publication in PLOS Pathogens.

Best regards,

Michael Malim

Editor-in-Chief

PLOS Pathogens

orcid.org/0000-0002-7699-2064